# A mutational atlas for Parkin proteostasis

Lene Clausen[1,5], Vasileios Voutsinos [1,5], Matteo Cagiada[1],
Kristoffer E. Johansson [1], Martin Grønbæk-Thygesen[1], Snehal Nariya[2],
Rachel L. Powell [2], Magnus K. N. Have[1], Vibe H. Oestergaard [3],
Amelie Stein [3], Douglas M. Fowler [2,4] ✉, Kresten Lindorff-Larsen [1] ✉ &
Rasmus Hartmann-Petersen [1] ✉

Proteostasis can be disturbed by mutations affecting folding and stability of the encoded protein. An example is the ubiquitin ligase Parkin, where gene variants result in autosomal recessive Parkinsonism. To uncover the pathological mechanism and provide comprehensive genotype-phenotype information, variant abundance by massively parallel sequencing (VAMP-seq) is leveraged to quantify the abundance of Parkin variants in cultured human cells. The resulting mutational map, covering 9219 out of the 9300 possible single-site amino acid substitutions and nonsense Parkin variants, shows that most low abundance variants are proteasome targets and are located within the structured domains of the protein. Half of the known disease-linked variants are found at low abundance. Systematic mapping of degradation signals (degrons) reveals an exposed degron region proximal to the so-called "activation element". This work provides examples of how missense variants may cause degradation either via destabilization of the native protein, or by introducing local signals for degradation.

Parkinson's disease (PD) is an incurable neurodegenerative disorder that ensues from the loss of dopaminergic neurons in the *substantia nigra*, and is characterized by progressive loss of motor control, leading to tremor, rigidity, postural instability and bradykinesia[1]. In addition to sporadic PD, highly penetrant mutations in a few genes have been linked to familial PD. Of these monogenic PD forms, homozygous or compound heterozygous loss-of-function mutations in the *PRKN* (or *PARK2*) gene account for about 50% of autosomal recessive juvenile Parkinsonism (ARJP) (OMIM: 600116 [https://www.omim.org/entry/600116]) cases and 77% of familial early onset Parkinson's disease, starting at an age younger than 30 years[2,3].

The *PRKN* gene encodes the E3 ubiquitin-protein ligase, Parkin[4], which catalyzes protein ubiquitination and is required for mitophagy of damaged mitochondria[5,6]. The 465-residue protein is comprised of an N-terminal ubiquitin-like (UBL) domain, followed by a disordered linker, a really interesting new gene (RING) domain RING0, and the catalytic RING-in-between-RING (RBR) module, consisting of RING1, an

in-between RING (IBR) domain, a repressor (REP) element and finally the RING2 domain[7]. Within the disordered linker, a motif (residues 101-109) termed the activation element (ACT) contributes Parkin activation[8,9]. Biochemical and structural studies have shown that activation of Parkin requires a conformational change from an auto-inhibited state, where the UBL binds RING1 to block binding of the E2 ubiquitin-conjugating enzyme, while the RING0 domain directly occludes access to the catalytic cysteine residue at position 431 in RING2[10-13]. In response to damaged mitochondria, the kinase PINK1, encoded by another ARJP-linked gene, phosphorylates ubiquitin and Parkin[6,8,14,15]. In turn, this releases Parkin from its auto-inhibited state so that it can ubiquitinate nearby targets[16,17], including several mitochondrial outer-membrane proteins, ultimately leading to the clearance of damaged mitochondria by mitophagy[5,18]. Accordingly, mitochondrial dysfunction is recognized as a key event in both sporadic and familial PD. However, other non-mitochondrial Parkin targets have been identified[17], linking Parkin to degradation via the

[1]Linderstrøm-Lang Centre for Protein Science, Department of Biology, University of Copenhagen, Copenhagen, Denmark. [2]Department of Genome Sciences, University of Washington, Seattle, WA, USA. [3]Department of Biology, University of Copenhagen, Copenhagen, Denmark. [4]Department of Bioengineering, University of Washington, Seattle, WA, USA. [5]These authors contributed equally: Lene Clausen, Vasileios Voutsinos. ✉e-mail: dfowler@uw.edu; lindorff@bio.ku.dk; rhpetersen@bio.ku.dk

ubiquitin-proteasome system (UPS). Parkin has been shown to auto-ubiquitinate, thus regulating its own degradation[19,20]. During basal conditions, auto-ubiquitination is prevented by the UBL domain and REP, retaining the protein in a closed auto-inhibited state. Pathogenic mutations in the UBL domain have been shown to disrupt the auto-inhibited state, causing the protein to be constitutively active[21].

The PD-linked *PRKN* variants are spread throughout the gene[7,22], and include missense, nonsense and frameshift mutations. As non-sense and frameshift mutations typically cause large changes to the encoded protein, their consequences are usually deleterious and easily predictable. In contrast, the consequences of missense mutations, where one amino acid is replaced with another, are often more difficult to predict[23], but represent many of the protein-coding variants recorded in gnomAD (345/520 = 66%)[24] and ClinVar (92/423 = 22%)[25] (Simple ClinVar, accessed April 27th 2023).

Previous studies have shown that many disease-linked missense variants affect the folding and thermodynamic stability of the encoded protein[23,26,27]. In turn, this affects proteostasis, since the protein may be targeted by the protein quality control (PQC) system for degradation[28] resulting in insufficient cellular amounts of the protein[29–32]. Recent estimates indicate that as much as 60% of pathogenic missense variants cause loss of protein abundance[33,34], and proteome-wide predictions of protein stability changes show that disease-causing missense variants are more destabilizing than benign variants[27]. Nevertheless, the relationship between protein stability and cellular protein turnover and abundance is complex[35,36], and missense variants may affect other properties than thermodynamic stability. This suggests that systematic mapping of variant abundance may both provide valuable diagnostic information, while also highlighting the mechanisms and sensitivity of the PQC and proteostasis networks.

In recent years, the rapid decrease in DNA sequencing cost has made genome sequencing a standard tool in medicine, and consequently an increasing number of gene variants, whose pathogenicity is unknown, are being discovered. Accordingly, in *PRKN* alone, more than 60 missense variants of uncertain significance are reported in the ClinVar database[25]. Testing the consequences of such variants is an important but painstaking effort.

Here, we attempt to shed further light on variant effects in Parkin, and provide mechanistic insight into how missense variants may perturb proteostasis. We use variant abundance by massively parallel sequencing (VAMP-seq)[31] to probe the effects of 9219 out of 9300 (99.1%) possible single-amino acid variants and nonsense *PRKN* variants in large multiplexed experiments. Parkin variant abundance correlates with biophysical computational models of Parkin thermodynamic stability changes. In total, 28% (2431 of 8756) of all the measured Parkin missense variants and 50% (6 of 12) of the known PD-linked variants are degraded substantially more than the wild-type protein. Together with mapping of the intrinsic degradation signals in Parkin, these data may aid our ability to predict disease and help future implementation of precision medicine for PD.

## Results
### A multiplexed assay for Parkin variant abundance
Inspired by previous studies, showing that as much as 50–75% of pathogenic genetic variants cause loss of protein abundance in the cell due to loss of thermodynamic stability in folding[33,34,37], we set out to perform a deep mutational scan of Parkin protein abundance. Specifically, we aimed at applying the variant abundance by massively parallel sequencing (VAMP-seq) method[31] to a site-saturated library of *PRKN* variants. Here, a barcoded library of *PRKN* variants, fused to GFP, is introduced into HEK293T cells by recombination at a specific landing pad locus downstream of a Tet-on promoter (Fig. 1A). As the plasmid does not include a promoter, any unintegrated plasmids are not expressed and single-copy expression is achieved. To normalize for cell-to-cell fluctuations in expression, mCherry is expressed from

an internal ribosomal entry site (IRES) in the same construct (Fig. 1A). Before recombination, the cells express the blue fluorescent protein BFP, the inducible caspase 9 (iCasp9), and the Blasticidin S deaminase, which are each separated by autocleaving 2 A peptides. As correct integration at the landing pad will displace the BFP-iCasp9-Blast[R] coding sequences[38], non-recombinant cells can be selected against with the drug AP1903 (Rimiducid), which results in rapid loss of iCasp9-positive cells through apoptosis. Thus, based on the GFP:mCherry ratio, cells can be sorted into different bins and the variants in each bin can be identified and quantified based on short-read Illumina sequencing of the barcodes.

To test the system, we first compared wild-type (WT) Parkin with the R42P disease-linked missense variant, which is known to be thermodynamically destabilized, rapidly degraded[39,40] and present at a reduced steady-state level[19]. As expected, fluorescence microscopy (Fig. 1B) and western blotting (Fig. 1C) revealed dramatically reduced levels of the R42P variant. Though this appeared independently of whether the GFP-tag was located in the N- or the C-terminus of Parkin (Fig. 1B), we continued with GFP in the N-terminus of Parkin, since the Parkin C-terminus is partially buried in the structure, and the N-terminal GFP also allows for analyses of nonsense variants.

We tested the function of the GFP-Parkin fusion by measuring its ability to induce mitophagy of the mt-Keima reporter[41,42] in cells treated with antimycin and oligomycin (AO) (Supplementary Fig. 1A). Indeed, this revealed that WT Parkin was active, while a catalytically dead (C431A) variant was not (Supplementary Fig. 1B, C). In agreement with previous reports[19,40,43], the disease-linked R42P variant also appeared functional (Supplementary Fig. 1C). This is likely due to R42P being overexpressed in our system (Supplementary Fig. 1D) and suggests that the pathogenicity of the R42P variant is linked to its low abundance. As we were unable to detect endogenous Parkin in the cells (Supplementary Fig. 1D), measurement of the cellular abundance of recombinant Parkin variants is likely independent of endogenous Parkin. Flow cytometry measurements showed that the R42P level was reduced approximately 10-fold compared to WT (Fig. 1DE), which provides a sufficient dynamic range for VAMP-seq.

### A saturated map of Parkin variant abundance
We used VAMP-seq to determine the steady-state level of thousands of Parkin missense and nonsense variants. To this end, a site-saturated library of *PRKN* variants was inserted into the expression vector in frame with GFP (Fig. 1A). The *PRKN* library was recombined into the landing pad in the HEK293T cell line and non-recombinant cells were eliminated by treating with AP1903. The vast majority of Parkin library variants displayed GFP and mCherry levels similar to WT Parkin, whereas a smaller population displayed lower GFP levels overlapping with R42P Parkin (Fig. 1D). Accordingly, the majority of Parkin library variants were covered in the range between WT and R42P Parkin. Fluorescence-activated cell sorting (FACS) was used to sort the cells into four separate bins according to their GFP:mCherry ratio (Fig. 1F), and Illumina sequencing of the barcodes was performed to quantify the frequency of each variant in each of the four bins. Finally, for each variant we calculated an abundance score with 1 indicating WT-like abundance and 0 representing strongly reduced abundance. The scores and standard deviations were determined from four biological replicates each with three FACS replicates. These scores correlated well between replicate experiments (all Pearson's correlations were in the range of 0.96 to 0.99) (Supplementary Fig. 2). The resulting dataset displays the relative abundance of 8757 out of 8836 (465 residues x 19 amino acid substitutions per position + 1 wild-type) possible single amino acid variants and 462 of 464 (465-1 positions for early stop codons) nonsense variants corresponding to 99.1% and 99.3% coverage, respectively (Fig. 2A). Only three positions, 1, 344, and 345, were missing more than three substitutions and the majority of these were missing due to failure during the library synthesis. The distribution of

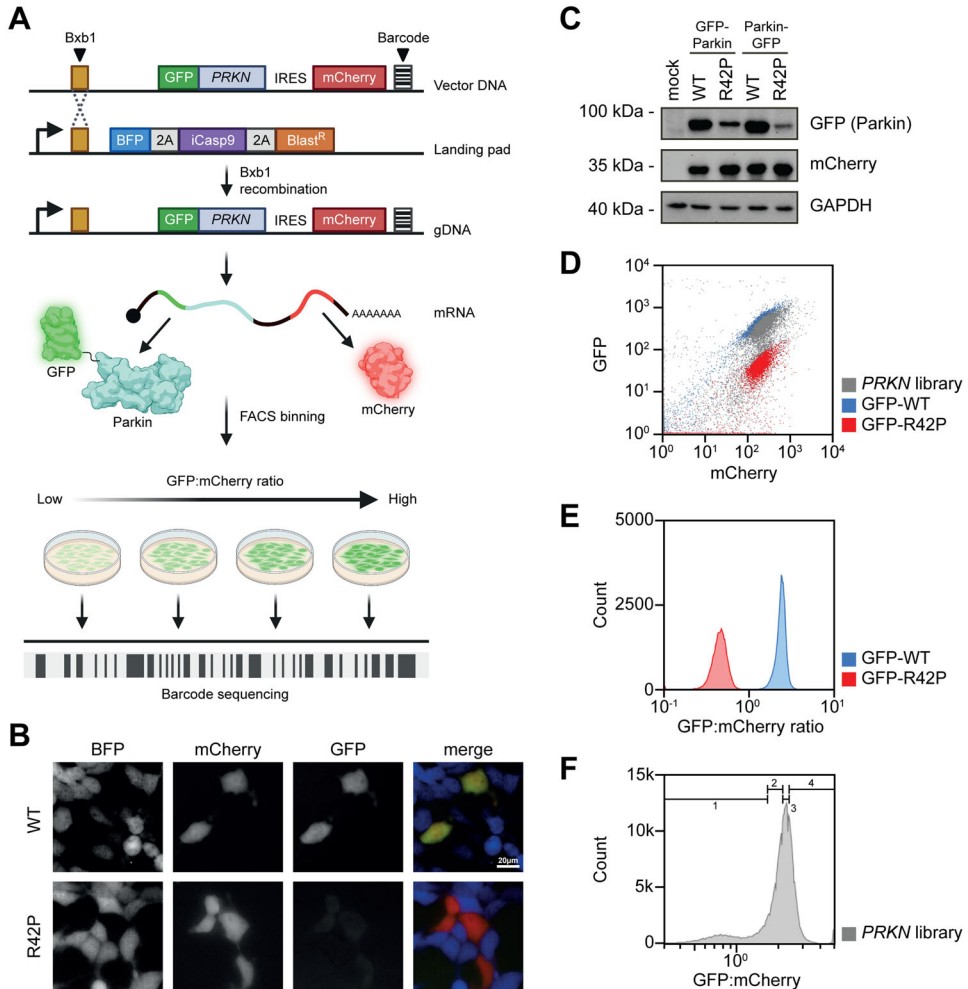

**Fig. 1 | Assessment of Parkin variant abundance. A** Schematic illustration of VAMP-seq applied to a site-saturated library of *PRKN* variants. The *PRKN* plasmid DNA library (Vector DNA) comprises a Bxb1-specific recombination site, a *PRKN* variant fused to GFP, an internal ribosomal entry site (IRES), mCherry and a unique barcode. The barcoded *PRKN* library is introduced into HEK293T cells containing a landing pad locus. The landing pad contains a Bxb1 recombination site downstream of a Tet-on promotor (bent arrow) that drives the expression of BFP, inducible Caspase 9 (iCasp9) and a blasticidin resistance gene (Blast^R) separated with a par-echovirus 2A-like translational stop-start sequence (2 A). The cells are co-transfected with the barcoded *PRKN* library and a plasmid encoding the Bxb1 recombinase that catalyzes site-specific recombination. After correct integration, GFP-Parkin and mCherry are expressed from the same mRNA. Using fluorescence-activated cell sorting (FACS) cells are sorted into four different and equally populated bins and the Parkin variants in each bin can be identified by sequencing the barcode. Figure created with BioRender.com. **B** Live fluorescence microscopy images showing BFP, GFP and mCherry signal intensities in stable transfected landing pad cells expressing N-terminal GFP-tagged wild-type (WT) or low abundance variant R42P Parkin. **C** The protein levels of GFP-tagged Parkin in whole cell lysates from stable transfected landing pad cell lines expressing N- or C-terminally GFP-tagged wild-type (WT) or R42P Parkin determined using SDS-PAGE and Western blotting with an antibody against GFP. The protein level of mCherry and GAPDH served as a control for cell-to-cell fluctuations and as a loading control, respectively. **D** Representative flow cytometry scatter plots of landing pad cells expressing GFP-WT (blue, *n* = 52,023), GFP-R42P (red, *n* = 52,245) or the *PRKN* variant library (grey, *n* = 5.46×10^5). **E** Representative histogram plots for stable trans-fected landing pad cells expressing N-terminal GFP-tagged wild-type Parkin (GFP-WT) or R42P (GFP-R42P). Each histogram was created from at least 46,500 cells. **F** A representative flow cytometry profile for landing pad cells expressing the *PRKN* library (grey, *n* = 4.95×10^5). Bin thresholds used to sort the library into four (1-4) equally populated bins (25% in each bin) are shown by black horizontal range gates.

the abundance scores was bimodal with a peak of synonymous (silent) variants overlapping with the WT peak, and a peak of nonsense variants that consistently displayed low scores (Fig. 2B). Additionally, a number of variants at positions in the linker region between the UBL and RING0 domains and in the extreme C-terminus, displayed abundance levels higher than WT (Fig. 2AB). The obtained abundance scores were consistent with the GFP:mCherry ratios obtained for 11 Parkin variants determined individually by flow cytometry in low throughput (Fig. 2C), as well as with the levels of 52 variants measured previously by fluorescence microscopy in U2OS cells (Supplementary Fig. 3)[43].

With a unique and high-quality map of abundance effects in Parkin, we next set out to understand the molecular origins of the observed effects. The median abundance score per position

(excluding nonsense variants) and the entire map has a Pearson correlation coefficient of 0.84 (so that the position-medians explain about 70% of the total variance in the abundance scores); this in turn highlights how the tolerance to substitutions depends strongly on the position in the protein (Fig. 2AD). As expected, the structured domains were in general more sensitive to mutations, in particular to proline residues (Fig. 2A). For the exposed β-strands in the RING0 domain, roughly every second residue, corresponding to those pointing inwards, were sensitive to mutations while those pointing outwards were more tolerant (Fig. 2A and Supplementary Fig. 4). The disordered loop regions (indicated by low AlphaFold pLDDT score) appeared largely tolerant to amino acid substitutions with the notable exception of positions 101-126 (Fig. 2A). Here, most substitutions of hydrophobic residues to hydrophilic residues led to increased Parkin abundance,

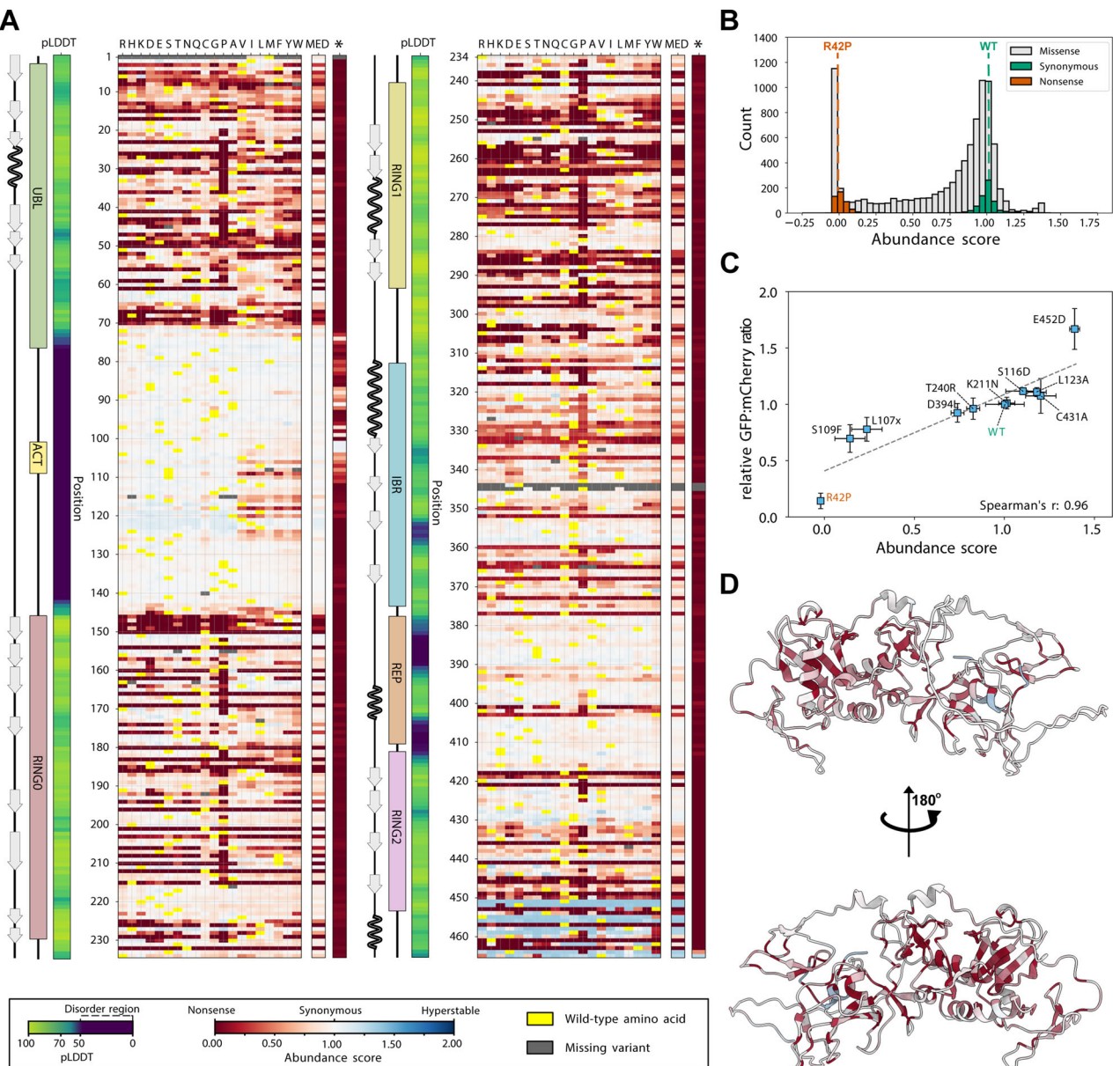

**Fig. 2 | A variant effect map of Parkin protein abundance. A** Heat-map of Parkin abundance scores determined by VAMP-seq. The asterisk (*) indicates nonsense variants. Median abundance scores (MED) based on missense variants were only calculated for the residues where we measured at least 17 abundance-scores. Scores range from low abundance (red) over WT-like abundance (white) to increased abundance (blue). Dark grey indicates missing variants. Yellow indicates the wild-type residue. We show the pLDDT (confidence score) from the AlphaFold prediction as a measure of the disorder propensity. The linear organization of Parkin domains and secondary structure elements across the sequence are indicated. **B** Histogram displaying the distribution of Parkin abundance scores for all missense (grey), synonymous (silent) wild-type (green) and nonsense (orange) variants determined by VAMP-seq. For comparison, the abundance scores of WT and R42P Parkin are marked with dashed lines. **C** Scatterplot comparing Parkin abundance scores derived from VAMP-seq and the GFP:mCherry ratios determined individually in low throughput by flow cytometry (*n* = 3 biological replicates). WT (green) and R42P (orange) Parkin were included for comparison. Error bars display standard deviations for both abundance scores and relative GFP:mCherry ratios determined in low throughput. x denotes a stop codon. **D** Cartoon representation of the Parkin structure (AF-O60260-F1) predicted by AlphaFold colored by the median abundance scores corresponding to panel **A**.

while exchanging hydrophilic to hydrophobic residues, led to a decreased abundance (Fig. 2A). As shown below, the latter effect is likely due to the introduction of a solvent-exposed degradation signal (degron).

To examine the variant effects structurally, we mapped the median abundance score at each position onto the Parkin structure (AF-O60260-F1, as predicted by AlphaFold [https://alphafold.ebi.ac.uk/entry/O60260]; Fig. 2D). The full-length Parkin crystal (PDB: 5C1Z, [https://www.rcsb.org/structure/5c1z])[12] and AlphaFold structures are similar (RMSD: 0.7 Å), but in the following we used the AlphaFold structure to enable visualization of the disordered loops. The

structural mapping of the abundance scores confirmed that the flexible regions and surface residues were, in general, tolerant to mutations, while the buried residues and those coordinating the $Zn^{2+}$ ions were highly sensitive (Fig. 2D and Supplementary Fig. 5). Accordingly, the median abundance score was generally high for positions that are exposed and thus have a low weighted contact number (WCN) (Supplementary Fig. 6A) or a high relative accessible surface area (rASA) (Supplementary Fig. 6B). When mapping only those positions with very low median abundance ( < 0.1) onto the Parkin structure, these appeared buried and were spread throughout the structured domains (Supplementary Fig. 7). This suggests that in most cases the low

abundance is caused by an underlying thermodynamic destabilization of the Parkin structure when missense variants are introduced at buried positions.

## Low abundance variants are thermolabile proteasome targets

We next sought to explore the molecular mechanisms that resulted in a low abundance of many variants. We first mapped the overall pathways of protein degradation by treating cells transfected with the *PRKN* library with either the proteasome inhibitor bortezomib (BZ) or chloroquine (CQ), which inhibits autophagy. The flow cytometry profile of the *PRKN* library showed that proteasome inhibition (Fig. 3A) shifted the peak of low abundance variants towards a higher GFP:mCherry ratio, an effect which was also observed upon knockdown of the proteasome subunit PSMD14 with siRNA (Fig. 3B). Conversely, no substantial changes were observed with chloroquine (Fig. 3C). Taken together, these experiments suggested that most low abundance variants are proteasome targets. As the thermodynamic folding stability of proteins is generally highly dependent on temperature, in particular for large proteins[44], the flow cytometry profiles were compared for cells incubated at 29, 37 or 39.5 °C (Fig. 3DE). At 39.5 °C the unstable peak became more pronounced, indicating that some of the variants with intermediate abundance are further destabilized (Fig. 3D). However, at 29 °C the low abundance peak almost disappeared entirely and most variants now appeared stable (Fig. 3E). Presumably this effect is the result of both an increased thermodynamic stability of the Parkin variants and a general reduction of protein turnover at the lowered temperature. Based on these results we suggest that many Parkin variants are thermolabile and most low-abundance variants are degraded by the proteasome.

As Parkin is an E3 enzyme we tested if the degradation was linked to Parkin-catalyzed auto-ubiquitination, as previously shown for the R42P variant[21,45]. Indeed, introducing the catalytically dead C431A substitution into WT or the R42P Parkin background led to a slightly increased abundance (Supplementary Fig. 8). Thus, it is possible that

some low abundance Parkin variants, in particular in the regulatory UBL domain, operate in this manner. We also note that hyperactive Parkin variants[43,46] tend to display a reduced abundance (Supplementary Fig. 9), which could be the result of increased auto-ubiquitination and/or a reduced structural stability. However, given that full activation of Parkin is a multi-step process and our experiments were all performed without inducing mitochondrial damage, most of the low abundance Parkin variants are likely subject to PQC-linked degradation due to an underlying destabilization of the native fold.

Small molecular stabilizers can restore protein abundance through binding and stabilizing the native structure of a protein[47]. Thus, we wanted to explore whether the small positive modulator of WT Parkin activity, BIO-2007817, discovered in a recent in vitro study[48] could confer stability to Parkin variants. However, treatment with the activator did not confer any substantial differences in Parkin abundance (Fig. 3F). Accordingly, we note that BIO-2007817 was also unsuccessful in increasing mitophagy in cell-based studies[48], and might only stabilize variants localized to the region or domain where the compound binds.

## Inherent degrons overlap with regions sensitive to mutation

When a destabilized protein is targeted for degradation, it is likely that the discriminating feature recognized by the degradation system is the exposure of degradation signals (degrons) through local or global unfolding events[23,28,49]. It has previously been shown that many such quality control degrons are enriched in hydrophobic residues and depleted for negatively charged residues[50–52].

To map degrons independently of the Parkin folding and stability, the full-length sequence was divided into 38 tiles of 24 residues, with each tile overlapping by 12 residues (Fig. 4A). These tiles were expressed fused to the C-terminus of GFP in place of the full-length Parkin variants in the VAMP-seq vector and cells were flow-sorted as before (Fig. 4B). Illumina sequencing of the tiles revealed the frequency of each tile in the four bins and was used to calculate a tile

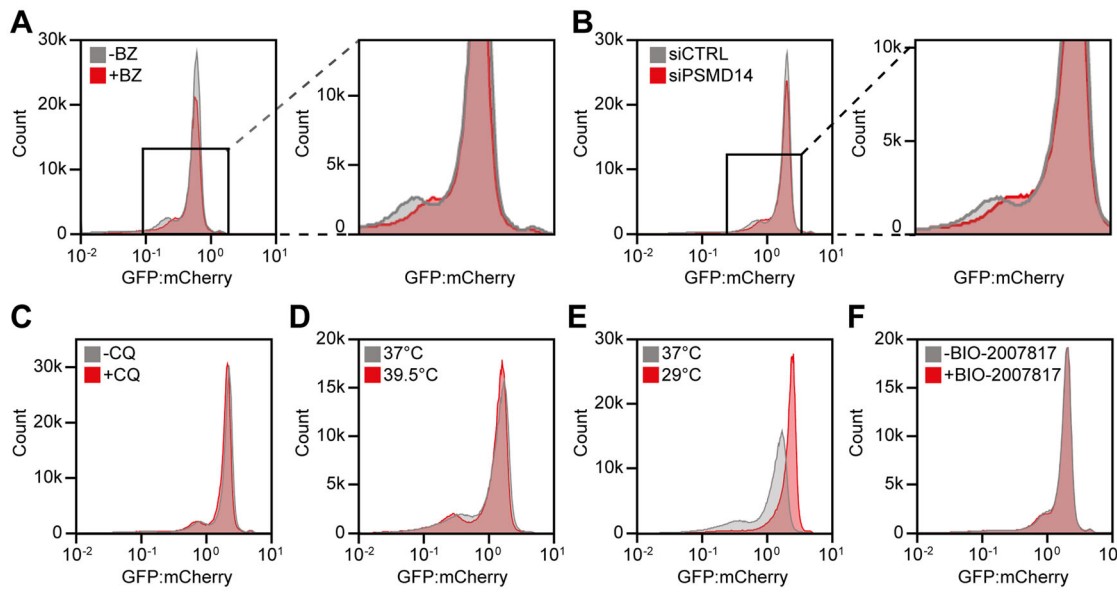

**Fig. 3 | The majority of low abundance Parkin variants are thermolabile proteasome targets.** Representative histograms for landing pad cells expressing the *PRKN* library untreated (-) (n = 559,000 cells) or treated (+) (n = 436,000 cells) with (**A**) 15 µM bortezomib (BZ) for 16 hours, (**B**) 25 nM siRNA against PSMD14 (siPSMD14) (n = 426,000 cells) or 25 nM control siRNA (siCTRL) (n = 506,000 cells) for 48 hours, or (**C**) without (-CQ) (n = 495,000 cells) or 20 µM chloroquine ( + CQ) (n = 516,000 cells) for 16 hours. Zoom ins, defined by black squares, are shown for

the BZ and siPSMD14 histograms. (DE) Flow cytometry profiles for landing pad cells expressing the *PRKN* library and grown at (**D**) 37 °C (n = 516,000 cells) or 39.5 °C (n = 511,000 cells) for 16 hours, or (**E**) 37 °C (n = 516,000 cells) or 29 °C (n = 511,000 cells) for 16 hours. **F** Representative flow cytometry profiles for landing pad cells expressing the *PRKN* library untreated (-) (n = 407,000 cells) or treated (+) (n = 426,000 cells) with 10 µM Parkin activator BIO-2007817 for 24 hours.

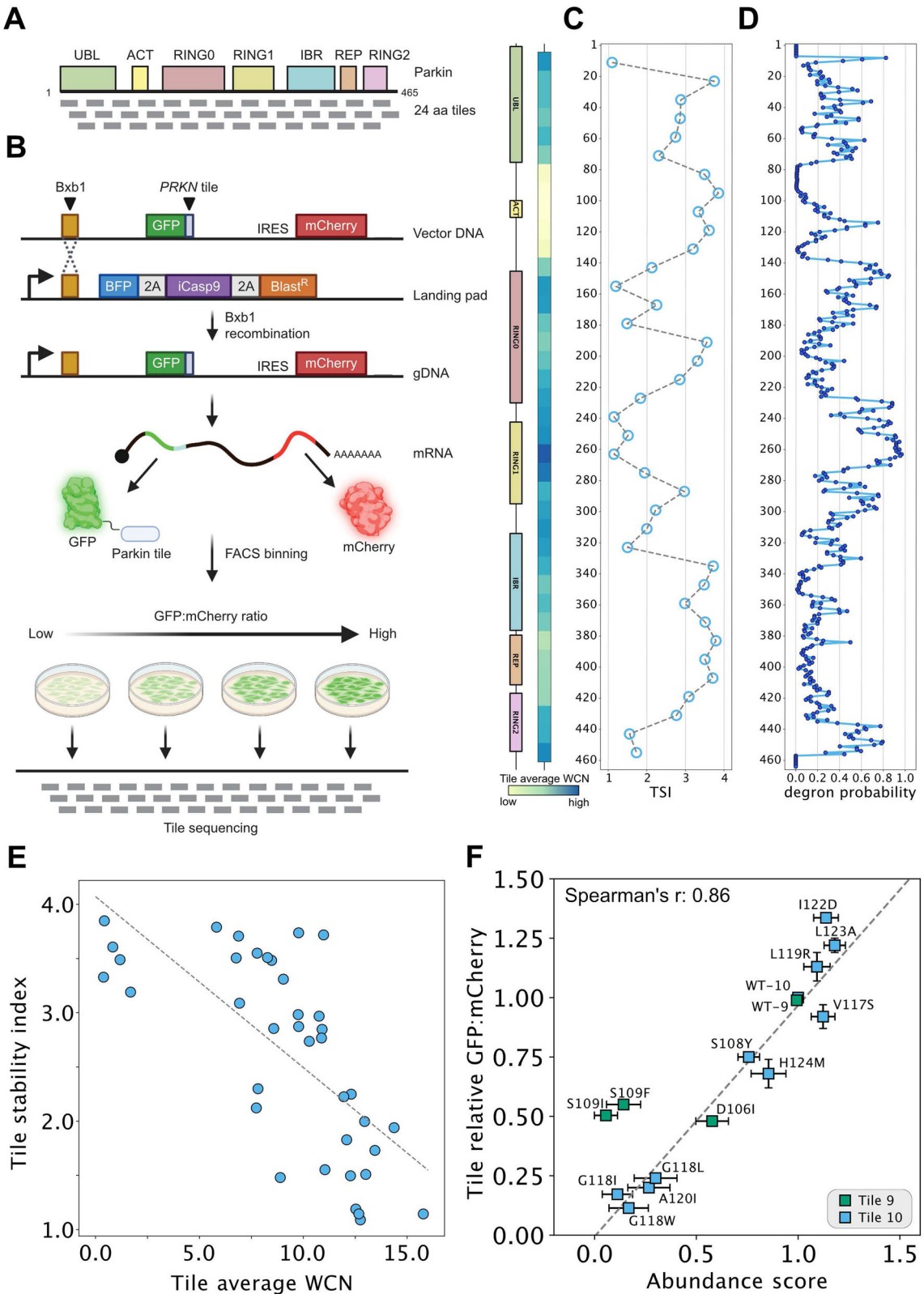

stability index (TSI) across the Parkin sequence (Fig. 4C). This revealed multiple regions with low stability index that caused a reduced abundance of the GFP fusion partner. Most of these tiles clustered to regions with structured domains (Fig. 4C), in line with the hydrophobic nature of the cores of the domains. To substantiate that the low abundance Parkin tiles act as quality control degrons, we applied the quality control degron predictor QCDPred[50] to determine the degron

probability across the Parkin sequence. This revealed that most low abundance tiles overlapped with regions with a high-quality control degron probability (Fig. 4D), confirming that these degrons contain the quality control degron features observed before[51]. Accordingly, when mapping the TSI onto the Parkin structure, most low-abundance tiles were found at buried positions (Supplementary Fig. 10), and the TSI correlated (Spearman's $r = 0.79$) with the extent of burial as

**Fig. 4 | Assessment of tile stability index to map degrons in Parkin. A** Schematic illustration of the linear architecture of Parkin divided into 38 tiles of 24 residues, each tile partially overlapping by 12 residues. **B** Schematic diagram of an adapted VAMP-seq approach applied to the 38 Parkin tiles as illustrated in panel A. Note that the approach is similar to that presented in Fig. 1A, except full-length Parkin variants were replaced with the 24-mer tiles. Moreover, the tile library is not barcoded hence tile sequencing is performed instead of barcode sequencing after sorting. Figure created with BioRender.com. **C** Tile stability index (TSI) assigned to the central position of each 24-mer tile. The two diagrams (left) show Parkin domains and average weighted contact numbers (WCN) with colors ramping from pale yellow (low WCN, corresponding to more exposed regions in the protein structure) to dark blue (high WCN, corresponding to more buried positions in the protein structure). **D** Predicted probabilities of 17-mer tiles being quality control degrons assigned to the central amino acid. **E** Scatterplot comparing the average WCN for each tile and TSI. Spearman's r: −0.76. **F** Scatterplot comparing GFP:mCherry ratios determined individually for Parkin tile missense variants in low-throughput by flow cytometry ($n = 3$ biological replicates) and abundance scores derived from the VAMP-seq experiment for full-length Parkin variants. Spearman's r: 0.86. The error bars indicate the standard deviation.

determined by the average weighted contact number (WCN) of the tile (Fig. 4E). Together, these results show that most degrons in Parkin are hydrophobic and buried inside the natively folded structure, and we suggest that these become exposed as the result of destabilizing missense variants.

For tiles in the 97-132 region (tiles 9 & 10) covering the ACT element in the flexible linker, we noted that this stretch displays a slightly reduced TSI, indicating that this fragment contains a weak degron. Since this entire region is embedded within the disordered linker between the UBL and RING0 domains (Fig. 2A), we reasoned that a degron at this position would be exposed also in the full-length protein, and observations made on full-length Parkin and the ACT degron tile should therefore correlate. To test this, we generated 13 variants in the context of these two tiles and measured their abundance by flow cytometry. Indeed, we find a strong correlation (Spearman's $r = 0.86$) between the abundance measurements of variants in the tiles and those obtained from our abundance map of full-length Parkin (Fig. 4F). Conversely, we did not observe any strong correlation for substitutions introduced in a buried tile (Supplementary Fig. 11). These results strongly suggest that variants in the disordered linker region do not cause low protein abundance via unfolding and exposing an existing degron, but rather enhance the inherent degron potential embedded within this exposed region of full-length Parkin. This is consistent with the abundance scores of missense substitutions where increasing or reducing the hydrophobicity, respectively strengthen or eliminate the degron activity (Fig. 2A) and the fact that the predictions support that the ACT region contains a quality control degron (Fig. 4D). The correlation between variant effects in tiles 9/10 and full-length Parkin suggests that degradation of these variants is not mediated by auto-ubiquitination of the full-length protein.

Finally, we noted that although the most C-terminal Parkin tile was of low abundance, QCDPred failed to detect this as a quality control degron (Fig. 4C, D). It is therefore possible that the low abundance of this fragment is due to Parkin carrying a C-degron[53] (which QCDPred was not trained to detect). Accordingly, the variant abundance map shows that truncation of the C-terminal V465 residue results in an increased Parkin abundance (Fig. 2A).

## Variant abundance correlates with stability and conservation

To further probe the PQC-linked degradation of Parkin variants, we computationally analyzed the thermodynamic stability and evolutionary conservation of all Parkin variants. We applied the Rosetta software[54] to estimate the thermodynamic change in folding stability compared to WT Parkin (ΔΔG). Substitutions that are predicted to preserve Parkin stability result in a ΔΔG close to zero, while positive values indicate a destabilization of the protein. Thus, the destabilized Parkin variants (ΔΔG predictions >0 kcal/mol) are expected to have a larger population of unfolded (or partially unfolded) structures that are targeted for degradation leading to reduced steady-state levels. The ΔΔG predictions reveal that 30% of the single amino acid substitutions are expected to change the Parkin stability by more than 2 kcal/mol, which previous studies indicate is sufficient to trigger degradation[55–57]. Overall, the thermodynamic stability predictions correlated relatively well with the Parkin abundance scores

(Spearman's $r = -0.46$ for all data, Spearman's $r = -0.55$ for residue median) (Fig. 5A), comparable to or slightly lower than those for other VAMP-seq experiments[55]. This indicates that the measured abundances are largely captured by thermodynamic stability predictions. Possible explanations for the imperfect correlation between predicted ΔΔG values and the abundance scores include (i) noise in the experimental measurements, (ii) the fact that Rosetta is an imperfect predictor of thermodynamic stability[58], and (iii) actual differences between changes in thermodynamic stability and cellular abundance. Examples of the latter might include specific effects of cellular quality control, or degradation via processes different from global unfolding. To examine one potential reason for differences between thermodynamic stability and cellular abundance, we focused on the 63 low abundance variants found at solvent-exposed positions and predicted not to perturb stability (ΔΔG < 1 kcal/mol). Of these we found that 28 variants (44%; mostly located in the ACT region but also in a small loop in RING0 and several aspartate residues in the UBL domain, Supplementary Fig. 12) were predicted to have a substantially increased quality control degron probability (Supplementary Fig. 12). This suggests that these variants (at already solvent-exposed positions) cause low abundance by degron formation rather than affecting thermodynamic stability. Side-by-side comparisons of the abundance and Rosetta ΔΔG maps are provided in the supplemental material (Supplementary Fig. 13), and suggest that loss of thermodynamic stability is nevertheless a major driver in degradation. In agreement with this suggestion, the abundance scores correlated with the melting temperatures of 35 Parkin variants analyzed previously in vitro (Supplementary Fig. 14A)[46]. While Rosetta did not accurately predict the destabilization of these variants (Supplementary Fig. 14B), we expect Rosetta will capture larger differences in Parkin stability.

As sequence conservation across species can predict the mutational tolerance of a protein at the residue level[59,60], we next analyzed the evolutionary conservation of all possible single residue variants using a Multiple Sequence Alignment (MSA) of 350 different Parkin homologues. We used the MSA as input to the GEMME software[61], which determines the evolutionary cost of introducing a given substitution in the wild-type sequence. This is achieved by combining analysis of the evolutionary relationships between the MSA sequences, the frequency of the single amino acid type at each position, and the possible epistatic relationship between residue pairs. The output is an evolutionary conservation score, which reports the likelihood of a given substitution at each sequence position. In our implementation, a large negative GEMME score indicates that the substitution is incompatible with the MSA and the variant should therefore be structurally unstable and/or non-functional. Conversely, a GEMME score close to zero indicates that the substitution is compatible with the MSA and should therefore be neutral (or WT-like in terms of function and stability). Indeed, GEMME has been shown to predict variant effects very accurately across a large set of MAVEs (https://www.proteingym.org/substitutions; accessed Nov 12, 2023) (Laine et al., 2019). As most proteins, Parkin need to be folded to function, sequence conservation across evolution contains a strong imprint of the protein structure and stability[62]. Indeed, the abundance scores overall correlated well with the GEMME scores (Spearman's $r = -0.60$) (Fig. 5B), and the correlation strengthened when comparing

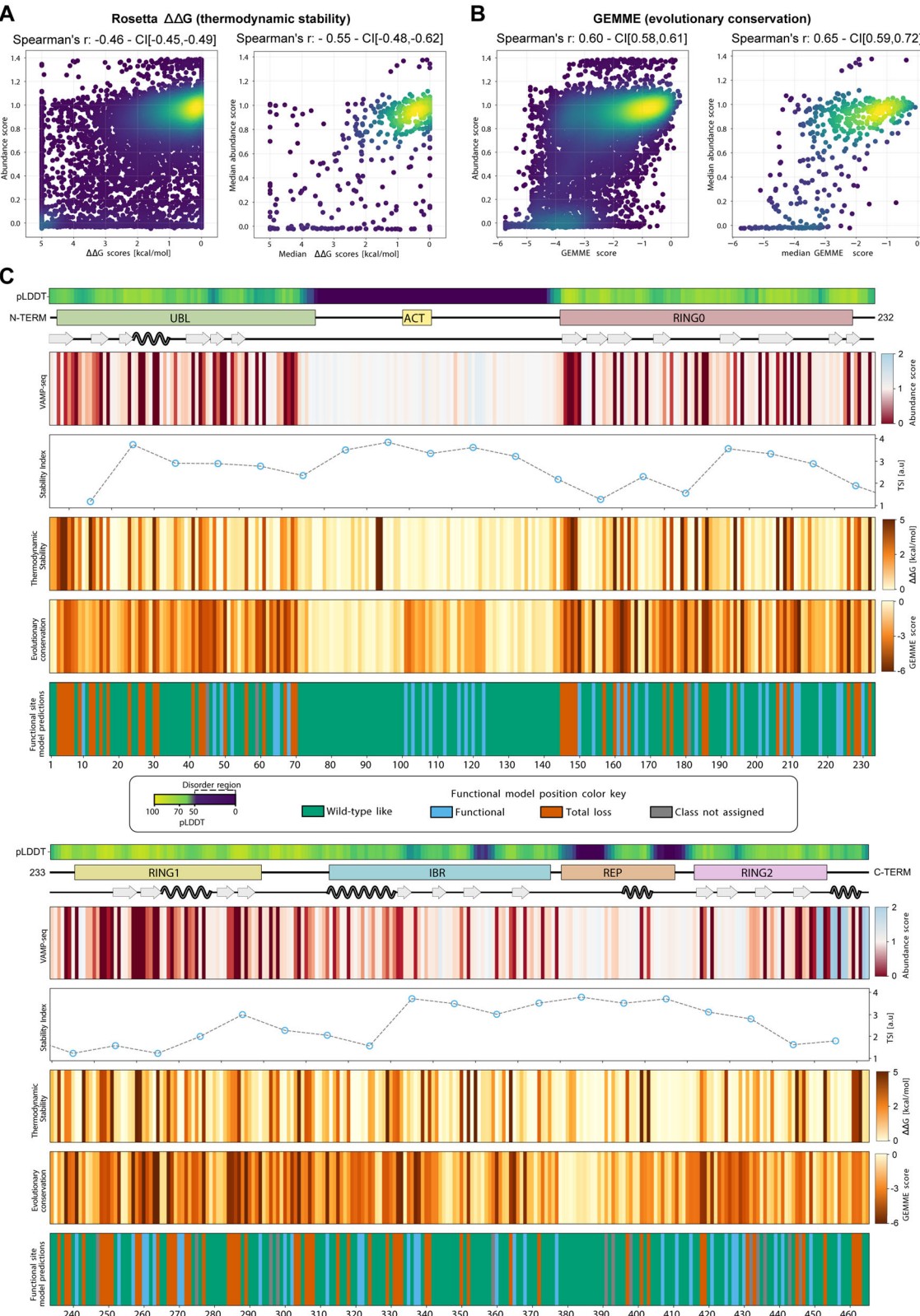

the residue median score (Spearman's $r = -0.65$). When comparing the abundance map with the GEMME map, it is noteworthy that the disordered region covering residues 101-126, where we see a reduced abundance for substitutions to hydrophobic residues and an increased abundance for substitutions to hydrophilic residues appears to be conserved. As GEMME scores are exclusively based on sequence conservation, this method cannot discriminate between residues that are

conserved for functional reasons, e.g. as those in the active site or at an interaction interface, or residues that are preserved to maintain a stable protein structure. Nevertheless, the results suggest that this exposed region plays a functional role, and that the hydrophobic nature and resulting degron potential is likely linked to this.

By combining the stability and conservation analysis, we can pinpoint those residues that are conserved for reasons beyond

**Fig. 5 | Parkin abundance scores correlate with thermodynamic stability and evolutionary conservation. A** Density scatter plots comparing all Rosetta ΔΔG scores and abundance scores for missense Parkin variants derived from VAMP-seq (left) or median Rosetta ΔΔG and abundance scores (right). Colors ramping from purple (low density) to yellow (high density). **B** Identical to panel A, except GEMME scores are compared to VAMP-seq-derived abundance scores. **C** The figure depicts three diagrams (top); displaying the AlphaFold confidence score, pLDDT, as a measure of predicted disorder, the linear organization of Parkin domains and motifs and secondary structure elements across the Parkin sequence. Furthermore,

the figure shows four maps and one line plot based on median scores; First, the abundance map and a line plot for tile stability index scores. Below the in silico maps based on evolutionary conservation scores by GEMME and thermodynamic stability ΔΔG values determined by Rosetta. Finally, a position effect map illustrating the classification obtained using a functional site predictor. Colors indicate the position classification. Green, wild-type-like positions (variants predicted as stable and functional); blue, functional sites (variants predicted as stable, but non-functional); red, total-loss sites (variants predicted as unstable and non-functional); grey, positions where a functional class could not be assigned.

maintaining protein stability, and we have recently shown that such analyses can be used to classify residues as being relevant for function or stability[63]. Using this approach, we assigned a class to each position based on the most common effect of the position's substitutions on structure and function (Fig. 5C). These functional classes are: 1) WT-like, i.e. positions where variants are predicted to be stable and functional. 2) Total-loss, where variants are predicted to be unstable and non-functional. 3) Functionally important sites, where variants are predicted to be stable, but non-functional. Overall, we find that 61 %, 21 % and 15 % of the positions in Parkin fall in these three classes, respectively, with the remaining 3% not showing any dominant class feature. While total-loss positions are mostly found in the buried region of the protein (81% with rASA <0.1) and are predominantly represented by hydrophobic residues, functionally important sites are mostly exposed on the protein surface (68% with rASA >0.1) and are positioned throughout the protein sequence, with only 5 of them closer than 10 Å to the active site C341 (Supplementary Fig. 15). Focusing on the positions that showed low abundance in the VAMP-seq data (<0.5 median abundance score at a position) we found that 61% were classified as total-loss positions, while 24% of the positions were predicted as functionally important sites, and the remaining positions were assigned as WT-like. For the conserved region overlapping with ACT, half of the positions classified as functionally important sites—in line with their conserved nature—while the rest classified as WT-like.

### Parkin abundance for identification of pathogenic variants
We next used our Parkin abundance data to examine Parkin variants that have already been observed in the human population. We first compared the Parkin variant effects with the allele frequency of the *PRKN* variants reported in the >140,000 human exomes available through the Genome Aggregation Database (gnomAD)[24]. This revealed that the most common *PRKN* missense alleles displayed WT-like scores (Fig. 6A), while several of the rare variants were of low abundance (Fig. 6A). Next, we collected a group of 12 disease-linked and 15 benign Parkin missense variants from the ClinVar database[64] of which 4 were initially reported as conflicting and then reclassified as benign or pathogenic by MDSgene[65] or according to the Sherloc criteria[43,66] (Supplementary Table 1). The benign variants all displayed abundance scores similar to WT, while half of the PD-linked variants (6 of 12) displayed abundance scores lower than 0.5 (Fig. 6B). The six disease-linked Parkin variants displaying WT-like abundance scores (close to 1) largely clustered around the active site (Supplementary Fig. 16), and four of these were predicted to be functionally-important sites (Source Data File). They are therefore likely pathogenic due to a loss of enzyme activity rather than through loss of protein stability. The ability of the abundance score to predict disease variants was analysed by a receiver operating characteristic (ROC) resulting in an area under the curve (AUC) of 0.69 (Fig. 6C). This is slightly lower than the predictive power of both the Rosetta ΔΔG and GEMME computational predictions (both have AUC = 0.79), which suggests that the abundance scores capture fewer aspects of Parkin pathogenicity. As expected, variant effect predictions made with EVE[67] were the most precise (Fig. 6C); the EVE scores are strongly correlated with the GEMME scores (Supplementary

Fig. 17), and EVE and GEMME scores correlate comparably with the abundance scores (Fig. 5 and Supplementary Fig. 17).

Finally, we used the abundance measurements and GEMME conservation scores to characterize the likely mechanisms of the disease-variants (Fig. 6D). As expected, most of the benign variants display WT-like abundance and conservation scores, with only a single variant having an abundance score <0.5 (R334C; score 0.48 ± 0.06). In line with the analyses above, we find that half of the disease-linked variants have low abundance. For the remaining six variants, four are evolutionarily conserved (GEMME < −2), indicating a connection to Parkin function (Fig. 6D), whereas two pathogenic variants (R33Q and P437L) both have GEMME and abundance scores that would normally indicate WT-like activity. Indeed, P437L is listed in ClinVar as having conflicting interpretations of pathogenicity, and it has a relatively high allele frequency in gnomAD ($1.6×10^{-3}$).

Functional class prediction on the six stable pathogenic variants marked one position as a total loss and the others as WT-like, and this approach is therefore not able to accurately capture these variants.

## Discussion
PD is, after Alzheimer's disease, the second most prevalent neurodegenerative disorder. Monogenic PD accounts for about 5% of all cases[68,69], of which *PRKN*-linked variants account for about half. Despite previous beliefs, it has become clear that genetics constitute a considerable proportion of the risk for PD. Hence, in addition to the rare monogenic forms of PD, common genetic variability at 90 loci is linked to an increased risk for the disease[70].

Given the known mutation rates for human cells, the size of the human genome and the global population, it is likely that every possible single nucleotide change compatible with life is found in the germline of the human population. Since the *PRKN* gene is located in one of the most fragile regions in the human genome, mutations here may be even more common[71]. The fragility of this genomic region may be tied to the large size of the *PRKN* gene[72] that covers 1.4 Mb, although the mature mRNA is only 4 kb. Importantly, the size of the region is conserved in vertebrates, suggesting an unknown regulatory function, but this comes at the cost of collisions between the transcription and replication machinery, and a resulting genomic fragility[71,73]. Of the *PRKN* variants listed (accessed April 27th 2023) in Simple ClinVar[25] 49% (208/423) have clinical interpretations, while the remaining variants (51%) as well as those not yet observed are of unknown significance. In addition, since Parkin may have tumor suppressor activity[74,75], somatic variants in the *PRKN* gene may also be relevant for cancer. Accordingly, characterizing the consequences of all presently known and unknown *PRKN* variants is of clinical value, but also provides information on Parkin structure and function, and the regulation of proteostasis.

Previous studies have analyzed the effect of PD-linked *PRKN* variants[40], and found that missense variants within the UBL domain decrease the stability of Parkin. For instance, in addition to the unfolding of the UBL domain, induced by R42P[39], this disables the UBL-RING1 interaction, rendering Parkin constitutively active[12], which in turn puts it at risk of auto-ubiquitination, resulting in reduced steady-state amounts. Our map indicates that many variants in the UBL domain cause destabilization, and may work in this manner. However,

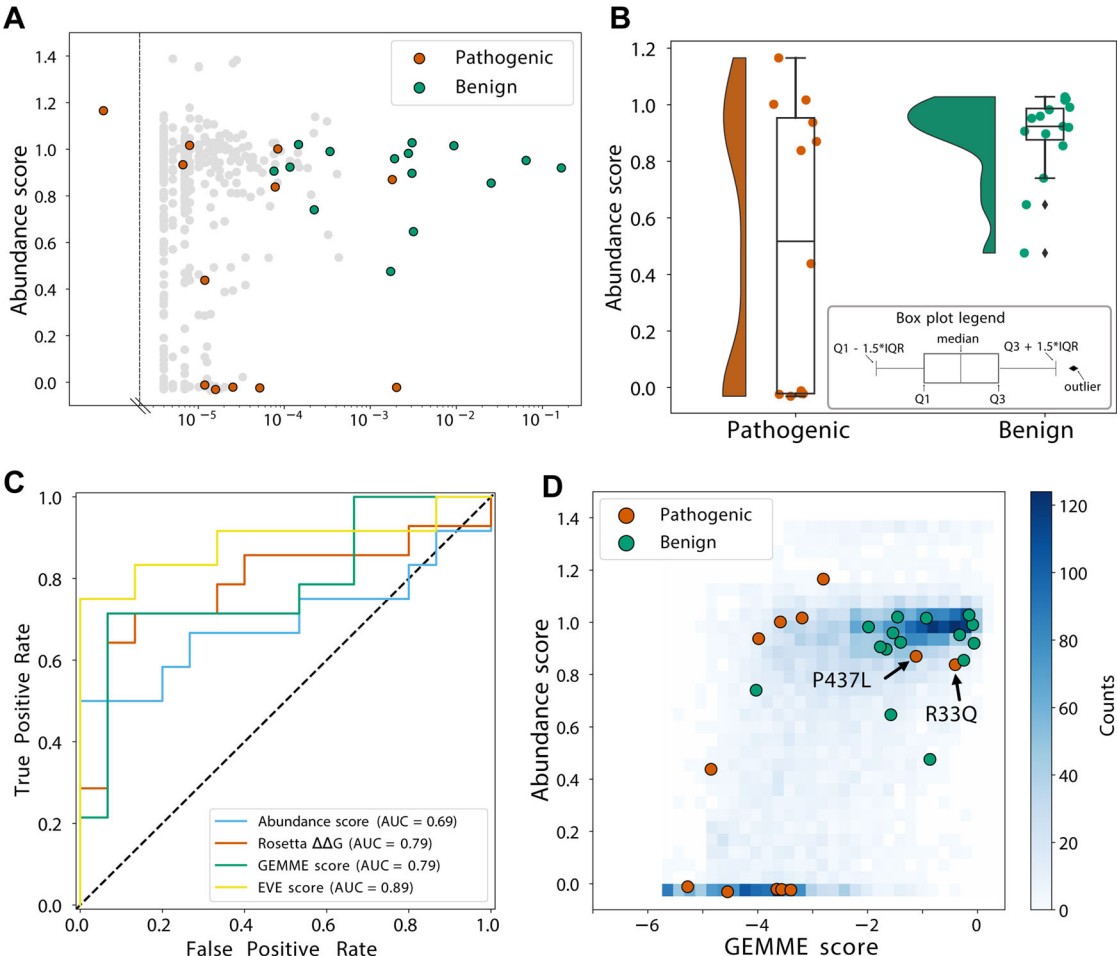

**Fig. 6 | Identifying and analyzing pathogenic variants. A** Scatterplot comparing abundance scores for missense Parkin variants generated by VAMP-seq and the allele frequency of pathogenic (orange), benign (green) or variants of unknown significance (grey) *PRKN* variants annotated in the Genome Aggregation Database (gnomAD). **B** Plots illustrate the abundance scores distribution for 12 pathogenic (orange) and 15 benign (green) Parkin variants. Dots are scattered horizontally to limit overlap, illustrating both the distribution and abundance score. The black diamonds indicate outliers. Boxplots are defined as indicated in the insert. **C** ROC curves for abundance score (blue), Rosetta ΔΔG (orange), GEMME score (green), and EVE scores (yellow). **D** Density scatterplot comparing GEMME scores and abundance scores for all variants (blue), variants classified as pathogenic (orange) or benign (green).

since the active site mutation C431A, only slightly stabilized R42P, any auto-ubiquitination effects are mild and the unfolding induced by R42P may also lead to its degradation via other E3 ligases through the PQC system. The observation that some Parkin variants that are hyperactive in vitro[46] also display a reduced abundance indicates that auto-ubiquitination may also contribute to the reduced abundance of these variants. It is likely, however, that most of the low-abundance Parkin variants are targets of the PQC system independent of auto-ubiquitination. In agreement with this, the abundance scores correlate with the predicted thermodynamic stability of the Parkin structure and with melting points determined in vitro[46]. In addition, several inherent PQC degrons are embedded within the Parkin sequence. These regions are buried in the context of folded, full-length Parkin, in line with the hydrophobic nature of PQC degrons[50–52,76,77]. In accordance with a recent report on an unrelated protein[49], we propose that for thermo-dynamically destabilized variants, buried PQC degrons become tran-siently exposed, leading to PQC-mediated proteasomal degradation of the Parkin protein variant.

Conversely, variants in or around the active site that lead to increased abundance, might do so due to Parkin loss of activity and in turn reduced auto-ubiquitination and degradation. However, since the C431A catalytically dead variant does not display a strongly increased abundance and given that Parkin activation is minimal in our assay,

another possibility, similar to the active sites of other enzymes[78–81], is that evolution at these positions has optimized functional properties of Parkin at the expense of thermodynamic stability. A similar situation may also hold for the degron region near and including the ACT ele-ment, where the hydrophobic ACT residues L102, V105 and L107 are required for Parkin activation[8]. Therefore the exposed region down-stream of the ACT region up to residue 126 may owe its increased hydrophobicity to intramolecular interactions that are necessary for Parkin activation. However, if so, this comes at the cost of rendering Parkin susceptible to degradation.

The full-length protein abundance correlates well with the equivalent tile variant abundance for the tiles that cover the area around the ACT element (residues 97–132). This is in agreement with the structural flexibility of this region, and suggests that missense variants at this position are likely to cause the formation of an exposed neo-degron, leading to a mechanism of degradation that is indepen-dent of thermodynamic stability of the Parkin structure. Potentially, certain substitutions of the aspartate residues in the UBL domain and variants in the short loop in RING0 may also generate exposed quality control degrons.

Because high abundance is a necessary, but not sufficient criterion for a variant to be functional, the abundance score is on its own not sufficiently powerful to separate all pathogenic and benign variants.

Thus, while low abundance variants are very likely to be pathogenic, we find that half of the known PD-linked missense variants have relatively high abundance scores (>0.5). The pathogenicity of most of these variants can be explained by either their function in thioester formation and ubiquitin ligation (T415N, G430D), or by the fact that they otherwise affect Parkin activation, such as in the case of K161N that leads to disruption of the interaction of RING0 with the phosphorylated UBL domain[43]. Similarly, G284R is thought to introduce major clashes with phosphorylated ubiquitin in the RING1 domain which would lead to decreased Parkin activity[43]. The mechanisms behind R33Q, M434T and P437L remain unclear; however, for the latter two their pathogenicity may be the result of their proximity to the active site.

Although recent technological advances, e.g. using CRISPR base editors[82], has made it possible to perform deep mutational scanning on genes at their endogenous locus, this is not readily compatible with high throughput measurements of protein abundance, especially in case of *PRKN* that is expressed at very low levels in most cell types, including as we show here in HEK293T cells. In the present study we therefore rely on overexpression of the Parkin variants. Though for some variants it is possible that the observed effects do not match up perfectly at endogenous expression levels, we note that our observations correlate with conservation, structural features and melting points, suggesting that our abundance read-out probes fundamental properties of Parkin thermodynamic stability which should be independent of the expression level. These results are also consistent with recent work that shows high correlations across deep mutational scanning experiments across different expression levels, with the main effect being a change in the dynamical range[83].

Our observation that half of the known PD-linked *PRKN* missense variants lead to low abundance, suggests that increasing the Parkin level may hold therapeutic potential for variants such as R42P, that albeit being destabilized and degraded is still functional[19]. Indeed, in line with these observations we also found that the pathogenic R42P variant can induce mitophagy when overexpressed. Presumably, at the low endogenous expression level, the additional reduction in abundance conferred by PQC-linked degradation of R42P, will result in an insufficient amount of Parkin. This highlights the advantage of the mechanistic insight gained from combining the variant abundance map with sequence conservation analyses. Hence, potentially blocking Parkin degradation or developing small molecule stabilizers or pharmacological chaperones, such as those for TP53 and CFTR[84,85], may restore cellular abundance and increase function above the pathogenic threshold. In principle, small molecules such as the allosteric Parkin modulator BIO-2007817 could increase the structural stability of Parkin through binding to its folded state. Although, we found that BIO-2007817 was unable to confer significantly increased Parkin variant abundance globally on the variant library, our cDNA library and the VAMP-seq approach may be useful for testing other small molecule Parkin binders in the future. Finally, as Parkin is also regulated on the transcriptional level[86] this may offer an orthogonal approach for increasing Parkin levels. In either case, such potential strategies will require detailed information on the molecular mechanism of the specific disease-linked variant. The variant effect map, provided here, is a first step towards such future precision medicine approaches to hereditary PD.

## Methods

### Site-saturation mutagenesis library cloning

A site-saturation mutagenesis library was ordered for *PRKN* (Twist Biosciences). The library was resuspended in 50 μL nuclease free water (Thermo Fisher Scientific, catalog numbers are provided in the source data file) individually (100 ng/μL). In a 50 μL reaction 1 μg of backbone plasmid (starting_eGPARK2iM) was digested at 37 °C for 1 hour with MluI-HF and EcoRI-HF (New England Biolabs), then heat-inactivated at 65 °C for 20 minutes. The digested products were purified on a 1%

agarose gel with 1x SYBR Safe (Thermo Fisher Scientific), followed by a gel extraction (Qiagen) and cleanup (Zymo Clean and Concentrate) of the 5.3 kb band following manufacturer's protocols. Using a Gibson reaction[87], the digested backbone product was assembled with the library oligonucleotide (diluted ten-fold) at a 2:1 molar ratio of insert:backbone at 50 °C for 1 hour. Then, assembly products were purified and eluted in 6 μL water (Zymo Clean and Concentrate).

To prepare for electroporation, 25 μL NEB-10β *E. coli* cells (New England BioLabs) were incubated with 1 μL of cleaned Gibson assembly product, digested backbone (no-insert negative control), or pUC19 (10 pg/μL). After a 30-minute incubation on ice, each sample was electroporated at 2 kV for 6 milliseconds. Cells were resuspended in 975 μL pre-warmed SOC media immediately after each electroporation. Each sample was incubated in a 37 °C shaking incubator (225 rpm) for 1 hour in a glass culture tube. Following recovery, each 1 mL culture was then added to its own 99 mL LB culture with 100 μg/mL ampicillin and grown overnight in a 37 °C shaking incubator. Prior to the overnight growth, 100 μL, 10 μL, and 1 μL samples from each flask were spread on LB-ampicillin plates to estimate library coverage by colony count. After the overnight growth at 37 °C, the Gibson-assembled product culture was centrifuged for 30 minutes at 4300 g and midiprepped (Millipore Sigma). Because of initial low library coverage, the library was reassembled and midi-prepped again, and the two preparations were combined before moving forward with barcoding.

### Barcoding of the site-saturation mutagenesis library

To barcode individual variants, 1 μg of library plasmid was digested at 37 °C for 5 hours with NdeI and SacI-HF (New England Biolabs). Subsequently, 1 μL rSAP (New England Biolabs) was added to the digested library product and incubated for 30 minutes at 37 °C, with a 20 minute heat-inactivation at 65 °C at the end. The digested library product was purified on a 1% agarose gel with 1x SYBR Safe (Thermo Fisher Scientific) then gel extracted (Qiagen). The digested library vector was purified and eluted in 10 μL water (Zymo Clean and Concentrate).

A barcoding oligonucleotide containing 18 degenerate nucleotides (IDT) was resuspended at 10 μM. To anneal the barcode oligo, 1 μL of the resuspended oligo was mixed with 1 μL 10 μM MAC356 primer, 4 μL CutSmart buffer (New England Biolabs), and 34 μL water. This reaction was run at 98 °C for 3 minutes, then ramped down to 25 °C at −0.1 °C/s. In order to fill in the barcode oligo, 1.35 μL of 1 mM dNTPs and 0.8 μL Klenow exo-polymerase (New England Biolabs) were added to the annealed product. This reaction was run at 25 °C for 15 minutes, 70 °C for 20 minutes, then ramped down to 37 °C at −0.1 °C/s. When the temperature reached 37 °C, the product was digested for 1 hour using 1 μL each of NdeI, SacI-HF, and CutSmart buffer. The digested product was then run on a 2% agarose gel with 1x SYBR Safe, gel extracted (Qiagen), cleaned and eluted in 30 μL water (Zymo Clean and Concentrate).

The library was ligated to the barcode oligonucleotide at a 7:1 oligo:library ratio overnight at 16 °C with T4 DNA ligase (New England Biolabs). A no-insert negative control was also ligated overnight using identical conditions. Following ligation, the products were individually purified and eluted in 6 μL water (Zymo Clean and Concentrate). NEB-10β *E. coli* were electroporated in a similar manner as described above with 1 μL of ligation product (including controls) or pUC19 (10 pg/μL). To bottleneck the library-barcode ligation product, electroporation recovery volumes of 500 μL, 250 μL, 125 μL, and 40 μL were added to different 50 mL LB-ampicillin cultures. For each of these 50 mL cultures, 100 μL, 10 μL, and 1 μL samples were taken and spread on LB-ampicillin plates to estimate library coverage. Following overnight growth at 37 °C, library coverage could be estimated by colony count. Following overnight growth at 37 °C, each of the 50 mL cultures were centrifuged for 30 minutes at 4,300 g and midiprepped (Millipore Sigma). The 250 μL bottlenecked library was used, resulting in an estimated 10x barcoded coverage of variants.

## Subassembly of the barcode-variant map by PacBio sequencing

The XmaI and NdeI enzymes (New England Biolabs) were used to digest 5 µg of each barcoded library in CutSmart buffer for 5 hours at 37 °C with a heat inactivation at 65 °C for 20 minutes at the end. Using AMPure PB beads (Pacific Biosciences), the digested products were purified. Both *PRKN* library preparation and DNA sequencing were performed at University of Washington PacBio Sequencing Services. At all steps, DNA quantity was checked with fluorometry on the DS-11 FX instrument (DeNovix) using the Qubit dsDNA HS Assay Kit (Thermo Fisher Scientific) and sizes were examined on a 2100 Bioanalyzer (Agilent Technologies) using the High Sensitivity DNA Kit. Preparation of the SMRTbell sequencing libraries was done according to the protocol 'Procedure & Checklist - Preparing SMRTbell Libraries using PacBio Barcoded Universal Primers for Multiplexing Amplicons' and the SMRTbell Express Template Prep Kit 2.0 (Pacific Biosciences). Then, in order to remove backbone fragments, SMRTbell libraries were size-selected using the SageELF (SageScience). The final library was bound with Sequencing Primer v4 and Sequel II Polymerase v2.1 and sequenced on a SMRT Cell 8Ms using Sequencing Plate v2.0, diffusion loading, 1 hour pre-extension, and 15-hour movie times. Circular consensus sequencing (CCS) reads were calculated using SMRT Link version 9.0 with default settings and reads having an estimated quality filter of ≥Q20 were designated as "HiFi" reads.

After library preparation, the barcoded library was pooled by normalizing mass to the number of constructs contained in the pool. The final library was bound with Sequencing Primer v4 and Sequel II Polymerase v2.0 and sequenced on two SMRT Cells 8 M using Sequencing Plate v2.0, diffusion loading, 1.5 hour pre-extension, and 30-hour movie time. Additional data were collected after treatment with SMRTbell Cleanup Kit v2 to remove imperfect and damaged templates, using Sequel Polymerase v2.2, adaptive loading with a target of 0.85, and a 1.3 hour pre-extension time. CCS consensus and demultiplexing were calculated using SMRT Link version 10.2 with default settings and reads having an estimated quality filter of ≥Q20 were selected as "HiFi" reads and used to map barcodes to variants.

Pacbio reads from the two sequencing runs were merged and aligned to the barcode-GFP-*PRKN* construct using BWA[88] and the barcode and *PRKN* sequences were extracted using cutadapt[89], see pacbio/pacbio_align.sh available at GitHub. Reads comprising ten or more DNA substitutions or any indels were removed and in cases where the same barcode mapped to more *PRKN* variants, the variant having most read counts was used to make a unique map of each barcode. This resulted in a barcode map of 257,610 unique barcodes, see pacbio/barcode_map.r. Of these, 15,974 are wild-type, 11,098 are synonymous wild-type, and 222,899 are single amino-acid variants including 5% nonsense variants corresponding to an average of 24 barcodes/variant. Nucleotide sequences are available on GitHub and summarized in supplementary material (Supplementary Fig. 18). More than 99% of all possible single amino acid substitutions are covered by this library and 445 of 465 positions are fully covered.

## Cell growth and maintenance

The HEK 293 T TetBxb1BFPiCasp9 Clone 12 cell line that was generated previously[38] was grown in Dulbecco's Modified Eagle's Medium (DMEM) supplemented with 10% v/v fetal bovine serum (FBS) (Sigma Aldrich), 0.24 mg/mL streptomycin sulphate (BioChemica), 0.29 mg/mL penicillin G potassium salt (BioChemica), 0.32 mg/mL L-glutamine (Sigma Aldrich) and 2 µg/mL doxycycline (Sigma-Aldrich). Cells were passaged when they reached 70-80% confluency and were detached with 0.25% trypsin (Gibco). The cells tested negative for mycoplasma. Authentication was performed by regular selection for recombinants with 10 nM of AP1903 (MedChemExpress) (see below) and checking for expression of BFP from the Tet-on promoter in non-recombinant cells.

## Integration of a single PRKN variant into the HEK293T landing pad

The cDNA of wild-type *PRKN* or *PRKN* variants studied in low-throughput were purchased from Genscript. Single *PRKN* variants were integrated into the Tet-on landing pad in the HEK 293T TetBxb1BFPiCasp9 Clone 12 cell line. First, $3.5 \times 10^6$ cells were seeded in 10 cm plates and left to grow overnight in media with no doxycycline. After 24 hours, 3 µg of the *PRKN* plasmid and 1 µg of pCAG-NLS-Bxb1 (9:1 molar ratio) were added in 400 µL of OptiMEM (Thermo Fisher Scientific). Then, 14 µL of Fugene HD (Promega) was added to the DNA/OptiMEM mixture before adding the entire transfection mix to the seeded cells. About 48 hours after this transfection 2 µg/mL doxycycline and 10 nM of AP1903 (MedChemExpress) was added to the cells, to activate gene expression from the Tet-on promoter and induce apoptosis in cells without recombination at the landing pad locus, respectively.

## Integration of the PRKN library into the HEK293T landing pad

The barcoded *PRKN* library was recombined into the Tet-on landing pad in the HEK 293 T TetBxb1BFPiCasp9 Clone 12 cell line. To this end, $3.5 \times 10^6$ cells were seeded in 10 cm plates and left to grow overnight in media with no doxycycline. After 24 hours, 7.1 µg of the *PRKN* library plasmid and 0.48 µg of pCAG-NLS-Bxb1 (17.5:1 molar ratio) were diluted with OptiMEM (Thermo Fisher Scientific) to a total volume of 710 µL in an Eppendorf tube. In a second Eppendorf tube, 28.5 µL of Fugene HD (Promega) was added to 685 µL OptiMEM (Thermo Fisher Scientific). The OptiMEM solution containing Fugene HD was added to the DNA/OptiMEM tube. The transfection mix was then added to the seeded cells in a 10 cm dish. About 48 hours, 2 µg/mL doxycycline and 10 nM of AP1903 (MedChemExpress) was added to the cells.

## Fluorescence live cell imaging

Recombined cells expressing either GFP-WT Parkin or GFP-R42P Parkin variant were seeded into a 96-well plate and imaged the next day using the InCell2200 cell imaging system (GE Healthcare). Fluorescence microscopy was performed using the excitation and emission filter settings of DAPI (excitation: 390 ± 18 nm, emission: 452 ± 48 nm) for BFP, FITC (excitation: 475 ± 28 nm, emission: 525 ± 48 nm) for GFP and TexasRed (excitation: 575 ± 25 nm, emission: 620 ± 30 nm) for mCherry.

## SDS-PAGE and western blotting

Cells were harvested in SDS sample buffer (SDS sample buffer (4x): 1.5% (w/v) SDS, 94 mM Tris/HCl pH 6.8, 20% glycerol, 0.01% Bromophenol blue, 1% (v/v) 2-mercaptoethanol). Samples were centrifuged and boiled for 2 minutes at 100 °C and run on 12.5% (w/v) acrylamide separation gels with a 3% (w/v) stacking gel. PageRuler prestained protein ladder (Thermo Fisher Scientific) was used as molecular weight marker. A constant voltage of 125 V was applied for approximately 1 hour, followed by blotting onto a 0.2 µm nitrocellulose membrane (Advantec) at 100 mAmp/gel for 1.5 hours. The areas of interest were cut out of the membrane and incubated in 5% (w/v) dry milk powder in PBS containing 0.1% (v/v) Tween-20 and 2.5 mM NaN₃ for at least 30 minutes. Next, primary antibodies were applied overnight. After extensive washing, horse radish peroxidase-conjugated secondary antibodies were applied for at least 1 hour. After extensive washing, the blots were dried with tissue paper and developed using Amersham ECL Western Blotting Detection system (GE Healthcare). Images were captured on a ChemiDoc Imaging System (BioRad). Antibodies and their sources were: rat anti-GFP (Chromotek, 3H9) (diluted 1:1000), mouse anti-mCherry (Chromotek, 6G6) (diluted 1:1000), mouse anti-Parkin (Santa Cruz Biotechnology, SC-32282) (diluted 1:1000), rabbit anti-GAPDH (Cell signaling Technology, 14C10) (diluted 1:1000).

## FACS profiling and cell sorting

Two days after transfection with recombination plasmids, cells were treated with 2 µg/mL doxycycline for 5-10 days before analytical flow cytometry or fluorescence-activated cell sorting. Cells were washed in PBS, trypsinized, resuspended in fresh media containing 2 µg/mL doxycycline, and seeded out while maintaining sufficient complexity for libraries (coverage of 100 cells/variant). Media and doxycycline was refreshed every third day. Perturbations were performed preceding FACS profiling as follows. Heat/cold: cells were moved to a separate incubator with the appropriate temperature for 16 hours prior to FACS profiling. Drugs: Cells were treated with 15 µM bortezomib (LC Laboratories) for 16 hours, 20 µM chloroquine (Sigma) for 16 hours or 10 µM Parkin activator BIO-2007817 kindly provided by Dr. Laura F. Silvian from Biogen. siRNA: Cells were reverse transfected for 48 hours prior to FACS profiling with 250 pmol (12.5 µL of a 20 µM stock) ON-TARGETplus Human PSMD14 siRNA (Dharmacon) or control siRNA (Dharmacon). The siRNA was mixed gently in 2 ml OptiMEM in a new 10 cm plate. Then. 20 µL Lipofectamine RNAiMAX (Invitrogen) was added and mixed gently followed by incubation for 10-20 min before $2 \times 10^6$ cells were seeded into the transfection mix. The media was changed 24 hours after reverse transfection.

On the day of FACS profiling or cell sorting, cells were washed in PBS, trypsinized, resuspended in media, centrifuged at 300 g for 3 minutes. The media was aspirated, and the cells were washed in PBS by centrifugation as before. Finally, the cells were resuspended in (5% v/v) bovine calf serum in PBS and filtered through a 50 µm polyethylene mesh filter into a 5 mL tube.

For FACS profiling, cells were analyzed for fluorescence with a BD FACSJazz machine using flow cytometry. BFP expressed from unrecombined cells was excited with a 405 nm laser. GFP or mCherry expressed from the recombined landing pad was excited by 488 nm laser and a 561 nm laser, respectively. Live cells were gated using forward and side scatter before successfully recombined cells were gated on BFP negativity and mCherry positivity (supplementary Fig. 19). The filters were 450/50 for BFP, 530/40 for GFP and 610/20 for mCherry.

For flow cytometry of the mt-Keima reporter, 2 µM antimycin A and 2 µM oligomycin (AO) and 1 µM bafilomycin A1 were added 4 hours prior to flow cytometry. The devices used were a BD Biosciences ARIA III FACS machine and an LSR Fortessa. The filters used were 442/46 or 431/28 (Fortessa) for BFP, 530/30 for GFP and 610/20 for neutral and acidic mt-Keima. The wavelength of the excitation lasers used were the 405 nm for BFP and for neutral mt-Keima, 488 nm for GFP and 561 nm for acidic mt-Keima. Live, single cells were gated by using the forward and side scatter and then recombinant cells were selected based on their lack of BFP, and expression of GFP (supplementary Fig. 19).

For cell sorting, cells were analyzed with a BD Biosciences ARIA III FACS machine equipped with a 70 µm nozzle. The laser used for excitation of BFP was 405 nm, for GFP 488 nm and for mCherry 562 nm for FACS AriaIII. The filters used were 442/46 for BFP, 530/30 for GFP and 615/20 for mCherry. First a population of live, single cells was gated by using the forward and side scatter and then recombinant cells were selected based on their lack of BFP, and expression of mCherry (supplementary Fig. 19). A histogram of the GFP:mCherry ratiometric parameter was established on the FACSDiva software and gates were set to separate the whole library into four equally populated bins based on the GFP:mCherry ratio. Thus, cells were sorted by BD Biosciences ARIA III based on the GFP:mCherry ratio into 4 tubes, each containing 25% of the Parkin variant population in the library. Each tube containing approximately 1.1 million sorted cells was centrifuged at 300 g for 3 min, supernatant was aspirated and cells were resuspended in fresh media containing doxycycline and transferred to a 6 cm dish. After three days, cells were transferred to a 10 cm dish with fresh media with doxycycline and allowed to grow for additional two days before collecting the cells by centrifugation at 300 g for 3 min and transferred

to an Eppendorf tube. Cells were spun down again and pellets were stored at −80 °C.

## Genomic DNA extraction

Genomic DNA was extracted from cells using the DNeasy blood & tissue Kit (Qiagen) following the manufacturer's protocol titled "Protocol: Purification of Total DNA from Animal Blood or Cells (Spin-Column Protocol)" except for the following differences: Addition of 20 µL RNase A (10 mg/mL) instead of the recommended 4 µL RNase A (100 mg/mL), prolongation of the 10-minute incubation at 56 °C to 30 minutes and finally, elution was performed using 100 µL nuclease-free water.

## Genomic DNA amplification, purification and quantification

For each bin, eight 50 µL PCR reactions were prepared with the following final concentrations: ~50 ng/µL genomic DNA, 1x Q5 High-Fidelity Mastermix (New England BioLabs) and 0.5 µM LC1020/LC1031 primers (Supplementary Table 2). The initial denaturation was performed at 98 °C for 30 sec; followed by 7 cycles of denaturation at 98 °C for 10 sec, annealing at 60 °C for 20 sec and extension at 72 °C for 10 sec; a final extension at 72 °C for 2 min and a 4 °C hold. Eight 50 µL PCR reactions were combined in a 1.5 mL Eppendorf tube and mixed with 320 µL Ampure XP beads (Beckman Coulter) (0.8:1 ratio) and left to incubate at room temperature for 5 min. Then, the tubes were applied to a magnetic stand and left at room temperature until beads and DNA fragments were pelleted completely. After allowing DNA and beads to bind, the supernatant was aspirated and the pellet was washed in 70% ethanol. The supernatant was aspirated and residual ethanol was allowed to evaporate completely. DNA was eluted by adding 21 µL of molecular grade water and vortexed. The tubes were pulse centrifuged and left to incubate at room temperature for 2 min before transferred to a magnetic stand for 2 minutes to separate beads from the eluted DNA. Then, 8 µL of the eluted DNA were transferred to a new PCR tube. A second PCR was performed to shorten the amplicon and add the P5 and P7 Illumina cluster-generating sequences. Here, 40% of the DNA elute was mixed with 2x Q5 High-Fidelity Mastermix, 10X SYBR green and the indexed forward PCR2_Fw primer and one of the indexed reverse primers JS_R at 0.5 µM each. The initial denaturation was performed at 98 °C for 30 sec; followed by 14 cycles of denaturation at 98 °C for 10 sec, annealing at 63 °C for 20 sec and extension at 72 °C for 15 sec; a final extension at 72 °C for 2 min and a 4 °C hold. Amplicons were run on a 2% agarose gel with 1x SYBR Safe (Thermo Fisher Scientific) and extracted using GeneJET gel extraction kit (Themo Scientific) following manufacturer's protocol. Extracted amplicons were quantified using the Qubit 2.0 Fluorometer (Invitrogen) with the Qubit dsDNA HS Assay Kit (Thermo Fisher Scientific).

## Sequencing and analysis

The prepared amplicon libraries were sequenced using a NextSeq 550 sequencer with a NextSeq 500/550 High Output v2.5 75 cycle kit (Illumina) with custom sequencing primers LC1040 and LC1041 for read 1 and read 2 (paired-end) while the indices were read with the primers LC1042 and ASPA_PARK2_index2_re for index 1 and index 2 respectively.

Illumina reads were cleaned for adapter sequences using cutadapt[89] and paired-end reads were joined using fastq-join from ea-utils[90], see illumina/call_zerotol_paired.sh available at GitHub. Only barcodes with a perfect match to the barcode map were counted, see illumina/merge_counts.r. Barcode counts for the 48 pairs of technical replicates have an average Pearson correlation of 0.82 (range 0.71-0.94) for the 257,610 unique barcodes and an average of 2.1 mill. matched reads (range 1.3-4.8 mill. reads) per technical replicate. Technical replicates of each FACS bin were merged and normalized to frequencies without pseudo counts. For each biological and FACS

replicas, a protein stability index (PSI) was calculated per barcode using:

$$PSI_b = \frac{\sum_g g \times f_{b,g}}{\sum_g f_{b,g}} \tag{1}$$

where $f_{b,g}$ is the frequency of barcode $b$ in FACS gate $g$. Barcode PSI's were averaged per amino acid variant, $i$, and finally, the PSI's of all replicates were averaged and normalized using:

$$\text{abundance score} = \frac{\text{PSI}_i - \text{PSI}_{\text{stop}}}{\text{PSI}_{\text{WT}} - \text{PSI}_{\text{stop}}} \tag{2}$$

where $\text{PSI}_{\text{WT}}$ corresponds to the PSI value of the wild-type amino acid sequence while $\text{PSI}_{\text{stop}}$ is the median PSI value of stop substitutions per amino acid residue, see illumina/abundance.r. The 12 biological and FACS replicates reproduced abundance scores well (Supplementary Fig. 2) and the standard deviation per variant is reported as an error estimate in the data file (Source Data File). The PSI measure gives a robust and direct number for a variant's position among the quartiles of the fluorescence distribution (bins) and, together with this normalization, facilitates a more direct comparison of scores between VAMP-seq experiments which typically use this calculation of scores[31].

### Degron cloning
For the degron analysis, the protein sequences of Parkin, along with six other proteins, were used to construct the protein tile library. DNA sequence optimization was performed using the IDT codon optimization tool. Then, the sequence was divided into 72 nt long oligonucleotides, each overlapping by 36 nt except for the C-terminal tiles, which may comprise a longer overlap. Template switching at the overlapping parts of the tiles may generate unwanted PCR products. To avoid this, the tiles were divided into odd tiles (Odds), even tiles (Evens) and C-terminal tiles (CT) based on the tile position in the tile series of each protein. To generate complementary overlaps for Gibson assembly cloning, two 30 nt long adaptors were attached to the 72 nt long oligonucleotide sequences, resulting in 132 nt long oligos. In parallel, the same adaptors were used for three 66 nt long control oligonucleotide sequences resulting in 126 nt long control oligonucleotide sequences. The 3 control oligonucleotide sequences were based on the 22-a-long APPY degron (-RLLL) sequence[76], and two variants hereof that mildly (-RAAA) or strongly (-DAAA) stabilize the APPY degron. The 132 nt long oligonucleotide sequences along with three controls were purchased from IDT as 3 separate libraries containing Odds (complexity = 93), Evens (complexity = 91) or CT (complexity = 10).

The oligonucleotide sequences were turned into double-stranded DNA and amplified by PCR using the primers VV3 and VV4. The initial denaturation was performed at 98 °C for 30 sec; followed by 2 cycles of denaturation at 98 °C for 10 sec, annealing at 69 °C for 30 sec, and extension at 72 °C for 10 sec; followed by a final 72 °C incubation for 2 min. The PCR product was run on a 2% agarose gel with 1x SYBR Safe (Thermo Fisher Scientific). Subsequently, the PCR product band was gel extracted using the GeneJet gel extraction kit (Thermo Scientific).

Using the primers VV1 and VV2, the attB-EGFP-PTEN-IRES-mCherry_562Bgl[31] vector backbone was linearized by inverse PCR. The reaction was performed using 5 ng of the vector DNA as template with the following program: Initial denaturation at 98 °C for 30 sec; followed by 30 cycles of denaturation at 98 °C for 5 sec, annealing at 69 °C for 30 sec, extension at 72 °C for 3 min and 40 sec; followed by a final 72 °C incubation for 5 min. Using the Zymo Research kit following the manufacturer's protocol, the PCR product was cleaned and concentrated. Subsequently, the PCR reaction was digested by DpnI (New England BioLabs) overnight and run on a 1% agarose gel with 1x SYBR

Safe (Thermo Fisher Scientific). The digested band was gel extracted using the GeneJet gel extraction kit (Thermo Scientific).

The double-stranded oligonucleotide sequences from all three libraries (Odds, Evens and CT) were assembled into the attB-EGFP-PTEN-IRES-mCherry_562Bgl linearized vector. This was done by Gibson assembly and mixing the oligonucleotide sequences with the vector in a 4:1 molar ratio. Then, the Gibson reaction was cleaned and concentrated with the Zymo Clean and Concentrator-5 kit. Subsequently, NEB 10-beta electrocompetent *E. coli* cells were transformed by electroporation with 2 kV. After that, the electroporated cells were incubated for 1 hour at 37 °C in 1 mL LB media. A 100 fold dilution was prepared of which 100 µL was plated on LB-ampicillin plates. The rest (900 µL) of the electroporated cells were transferred into 100 mL LB-ampicillin liquid cultures and incubated overnight. Then, plates were counted for colony formation units (CFU) to ensure a minimum of 100x of the complexity of each library and only then the 100 mL cultures were midi prepped (Millipore Sigma) and the DNA concentration was assessed by NanoDrop spectrometer ND-1000.

### Tile scoring
The tiles were integrated in the HEK 293 T TetBxb1BFPiCasp9 Clone 12 cell line with a similar approach as described for the *PRKN* variants. Like for the Parkin VAMP-seq experiment, tiles were sorted into 4 bins based on their GFP:mCherry ratio and DNA was extracted from the bins. Amplicons were prepared for downstream Illumina high-throughput sequencing with primers VV40S and VV2S. For the first PCR, initial denaturation was performed at 98 °C for 30 sec; followed by 7 cycles of denaturation at 98 °C for 10 sec, annealing at 65.5 °C for 10 sec and extension at 72 °C for 50 sec; followed by a final extension at 72 °C for 2 min. Ampure XP beads (0.8:1 ratio) was used to purify the PCR product before the Illumina cluster generation sequences were added with a second PCR with the primers gDNA_2nd and JS_R. For the second PCR, initial denaturation was done at 98 °C for 30 sec; followed by 16 cycles of denaturation at 98 °C for 10 sec, annealing at 63.5 °C for 10 sec and extension at 72 °C for 10 sec. Using a NextSeq 500/550 Mid Output v2.5 300 cycle kit (Illumina), the amplicons were sequenced using custom sequencing primers VV16 and VV18 for read 1 and read 2 (paired-end) and primers VV19 and VV21 for index 1 and index 2, respectively.

Identical to the processing of reads in the Parkin VAMP-seq experiment, the tile reads were cleaned for adapters sequences using cutadapt[89] and paired end reads were joined using fastq-join from eautils[90], see illumine_degron/call_zerotol_paired.sh available at GitHub. Only barcodes having a perfect match to the barcode map were counted, see illumine_degron/merge_counts.r. When tiles from the Odds, Evens or CT libraries were noticed in a sorting of a different library, these were considered to be non-sorted contaminants and disregarded. Technical replicates of each FACS bin were merged and normalized to frequencies having no pseudo counts. For each library, biological and technical/FACS replicates had a tile stability index (TSI) determined per tile using the equation:

$$TSI_t = \frac{\sum_g g \times f_{t,g}}{\sum_g f_{t,g}} \tag{3}$$

where $f_{t,g}$ corresponds to the frequency of tile $t$ in FACS gate $g$, see illumine_degron/tile_stability.r. Two of the APPY based control tiles, RLLL and DAAA, were observed in all sequenced pools and used to renormalize the Evens and CT libraries to equal the TSI of the control tiles from the Odds library using the equations:

$$TSI_t^{\text{even,norm}} = 0.075 + 0.9018 * TSI_t^{\text{even}} \tag{4}$$

$$TSI_t^{\text{ct,norm}} = 0.5895 + 0.5570 * TSI_t^{\text{ct}} \tag{5}$$

The complexity of the libraries is relatively low. Consequently, each tile was on average covered by more than 3500 detected reads per technical replicate. Thus 3 biological and 2 technical/FACS replicates for each of the 3 libraries generated TSI scores with a minimum Pearson correlation of 0.97. The standard deviation per tile is noted as an error estimate in the source data file.

## Evolutionary conservation scores

We calculated the "evolutionary distance" for Parkin isoform-1 for all the variants using sequence conservation information. We first retrieved Parkin homologs and generated a multiple sequence alignment using HHblits[91] with an E-value threshold of $10^{-20}$. The raw sequence alignment included 1379 sequences, which we reduced to 350 homologs by filtering out sequences having more than 50% gaps. Finally, we determined evolutionary conservation scores using this filtered sequence alignment as input to the Global Epistatic Model for predicting Mutational Effects (GEMME) software[61].

EVE scores[67] were obtained from the EVE webpage (https://evemodel.org/).

## Thermodynamic stability predictions

All structural analyses and visualizations were based on a structural model predicted by Alphafold2 available from www.alphafold.ebi.ac.uk under accession AF-O60260-F1[92] with zinc ions added using AlphaFill[93]. The predicted structure has the advantage of containing all residues and previous work have shown that AF2 models in general performs like experimental structures for Rosetta stability calculations[94]. Disordered regions are assigned based on the pLDDT confidence score from AlphaFold[94]. We performed predictions of changes in thermodynamic stability (ΔΔG) using Rosetta (GitHub SHA1 99d33ec59ce9f-cecc5e4f3800c778a54afdf8504) with the Cartesian ddG protocol[54]. All the ΔΔG values obtained from Rosetta were divided by 2.9 to convert them to kcal/mol[54].

## Statistics and Reproducibility

The VAMP seq. screening was performed 12 times in total: 4 biological repeats (separate library transfections/selections B1-4), with 3 repeats of the cell sorting for each (FACS1-3). At all steps we made sure to maintain 100-fold coverage of the library complexity (i.e. at least 9300 variants x 100 = 930,000). For the tile sequencing we performed 3 biological replicates (separate library transfections/selections), with 2 repeats of the cell sorting for each. We aimed for a 1000-fold coverage of the library complexity (i.e. at least 194 ×1000 = 194,000). All other experiments were performed at least three times with similar results.

## Reporting summary

Further information on research design is available in the Nature Portfolio Reporting Summary linked to this article.

## Data availability

The sequencing data generated in this study have been deposited in GEO under accession code GSE254618 and GitHub under accession code https://doi.org/10.5281/zenodo.8009574 [https://github.com/KULL-Centre/_2023_Clausen_parkin_MAVE]. Abundance scores are also deposited at MaveDB: entry urn:mavedb:00000114 [https://www.mavedb.org/#/experiments/urn:mavedb:00000114-a]. Sequencing reads for the abundance scores are available at https://doi.org/10.17894/ucph.ef2e30c5-d262-4713-86e8-a3964b5dd6c7 and for the degron scores https://doi.org/10.17894/ucph.d879cfce-efb3-4eaa-928f-87a94d9560ef. All the data are available freely. In addition, data was used from MDSGene (https://www.mdsgene.org/), ClinVar (https://www.ncbi.nlm.nih.gov/clinvar/), and EVE (https://evemodel.org/). The processed data are available in the source data file provided with this paper. Source data are provided with this paper.

## Code availability

All software generated for this article is available on GitHub: https://github.com/KULL-Centre/_2023_Clausen_parkin_MAVE (https://doi.org/10.5281/zenodo.8009574).

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

## Acknowledgements

We acknowledge the use of the FACS and computing core facilities at the Biotech Research & Innovation Centre and Department of Biology, University of Copenhagen. We thank Michael Lisby, Søren Lindemose and Anne-Marie Lauridsen for assistance. We thank Nicholas A. Popp for assistance with the cloning strategy and methods. The allosteric Parkin modulator was kindly provided by Dr. Laura F. Silvian from Biogen. The present work was funded by the Novo Nordisk Foundation (https://novonordiskfonden.dk) challenge program PRISM (to K.L.-L., A.S., D.M.F. & R.H.-P.), the Lundbeck Foundation (https://www.lundbeckfonden.com) R272-2017-452 and R209-2015-3283 (to A.S.) and R249-2017-510 (to L.C.), and Danish Council for Independent Research (Det Frie Forskningsråd) (https://dff.dk) 10.46540/2032-00007B (to R.H.P.).

## Author contributions

L.C., V.V., M.C., K.E.J., M.G.-T., V.H.O., S.N., R.L.P., M.K.N.H. and A.S. performed the experiments. L.C., V.V., M.C., K.E.J., A.S., D.M.F., K.L.-L. and R.H.-P. analyzed the data. D.M.F., K.L.-L. and R.H.-P. conceived the study. L.C., V.V and R.H.-P. wrote the paper.

## Competing interests

K.L.-L. holds stock options in and is a consultant for Peptone Ltd. The remaining authors declare no competing interest.
