## [Peer Review File · Nature Communications]

REVIEWER COMMENTS

Reviewer #1 (Remarks to the Author):

It is established that numerous types of parkin mutation cause parkin-related PD. Recently, several studies have indicated that many disease-linked point mutants of Parkin retain substantial catalytic activity suggesting other mechanisms contributing to loss of protein function and hence disease. Here, in this very interesting piece of work by Clausen et al, the authors aimed to understand how mutations in Parkin contribute to its dysfunction, by systematically defining destabilizing Parkin mutations. Given the link between PRKN and PD, this has therapeutic application as increasing Parkin protein levels by inhibiting its degradation may restore the protein function and reverse the disease.

The authors generated a comprehensive Parkin variant library and used Vamp-seq to assign stability scores to every possible variant. Although most mutation does not affect protein stability, some mutations showed significant lower stability. Experiments were complemented by computational approaches to calculate thermodynamic stability and evolutionary conservation in order to pinpoint residues that are preserved to maintain a stable protein structure. Altogether, the authors suggest that destabilizing Parkin mutations expose PQC degrons that lead to Parkin proteolysis. To map PQC degrons that might be exposed because of the Parkin mutation the authors screened a tiled peptide library of Parkin.

-Overall, the authors have done a good job of identifying destabilizing Parkin mutations and potential PQC degron. The manuscript writing is clear and concise and figures are presented nicely. This resource should be of great interest to the readership of Nature communications both for investigators from the UPS and proteostasis field as well as for wider scientific community.

My main concern is that authors did not assess the functional consequence of the destabilizing Parkin mutations; some of the destabilized mutants may still be functional, even though they are partially degraded. The E3 ligase activity, localization and biological function (e.g mitophagy) were not analyzed for low abundance Parkin mutants and as such it is not clear whether the reduced abundance overall affects the protein function.

Major points:

1. Most PRKN variants' stability is similar to WT, while a small bump of unstable population exists. PSI calculation to my understanding does not take into account the fact that FACS bins are not sized equally and the PSI difference between bin1 to 2 is not the same as bin2 to 3. Can authors comment on that?

2. Following up on the previous comment- can the authors create a figure similar to Fig S12 for variants enriched in bin1 only (those with PSIs 1-2, excluding stop mutants)? I wonder if the very low abundant variants cluster in specific protein domain or spread everywhere.

3. Authors need to show that reduced Parkin levels lead to reduced protein activity in biological context- mitophagy assays and/or E3 ligase activity. Otherwise, correlation between destabilizing mutations and disease cannot be made. It could be that reducing the protein levels by 10-fold does not have not a major impact on cellular functions related to it. As there are only several rare pathogenic disease associated variants of low abundance (Fig 6A), proof of concept can be done on those.

4. Expression from landing pad seems to be extremely high (Fig S1). To validate the Vamp-seq screens findings and exclude artifacts from overexpression and GFP-fusion, a few representative mutants need to be expressed under a weak promoter and protein levels need to be assessed in western blot. Parkin was reported be expressed endogenously in HEK293 cells so I would recommend comparing endogenous Parkin to exogenously variants in these cells.

5. I am not sure what is the added value of the peptide screen. Hydrophobic degrons were discovered, however, we do not know which of them is exposed and serves as a PQC degron in the context of Parkin low abundance variants. Do any of the Parkin variants is predicted to show severe structural changes and exposure of one of the discovered PQC degrons? Can authors provide an example? I am looking for a link between Parkin variant and peptide screens.

Reviewer #2 (Remarks to the Author):

The authors present the generation and assaying of over 9000 single site mutants of the Parkin protein using the variant abundance by massively parallel sequencing approach. Using this method, they quantified the amount of Parkin, a ubiquitin-protein ligase implicated in Parkinson's disease, in cultured human cells and then did several analyses to get at the role of Parkin stability in proteostasis, mitophagy and the pathology of Parkinson's disease. They found that the abundance of Parkin in cells correlates with the thermostability of the variant as well as its propensity to be degraded. Abundance also correlated with Evolutionary Conservation according to the GEMME metric, and ½ of the disease-linked Parkin variants have low abundance. Nevertheless, the best metrics for predicting the disease-linked mutants were Rosetta $\Delta\Delta G$ and GEMME score, which the authors interpret to mean that the abundance score only captures some of the pathogenic attributes of Parkin variants.

Overall, the work is extensive enough and of broad enough interest both from a basic science and an applied science standpoint to warrant publication in a journal targeted to a wide audience. The report is also well-written and the conclusions mostly justified. There are only a few problems they could address

to improve its presentation and support for their arguments. With these issues satisfactorily addressed, I see no reason not to publish these findings.

- The analysis using GEMME is being discussed as if it is a black box that everyone knows what is going on in there. Their introduction to the method didn't specify that "evolutionary conservation" refers to a metric derived from a global model that considers the role of epistasis. The background on this method needs to be explained better.

-Next, figure S10 doesn't depict anything about low conservation based on the evolutionary conservation score. The statement "When comparing the abundance map with the GEMME map, regions with low conservation also generally appeared tolerant to mutations (Fig. S10)" is really hard to see in figure S10. Perhaps there is another way to demonstrate this point or to annotate the current figure or preferably to come up with a numerical analysis instead.

-I have a problem with putting experimental data of thermostability in the supplemental (Fig. S11) instead of with the Rosetta analyses in the main text (Fig. 5). They should discuss the experimental thermostability data first and then the Rosetta predictions as a support, rather than the other way around. You should show the correlation of Rosetta $\Delta\Delta G$'s with the experimental ones from the 35 that were tested, and use that to bolster trust that Rosetta performs well in this system.

-“For the remaining six variants, four are evolutionary [sic] conserved($\Delta\Delta E < -2$), indicating a connection to Parkin function.”

This statement claims that evolutionary conservation increases the smaller the $\Delta\Delta E$ value is. But authors state on page 14, “Using GEMME, we calculated an evolutionary conservation score, where a score of zero predicts a WT-like variant, while variants that are predicted as unfavorable for structure and/or function result in a negative score.” I am confused as to how these two statements are consistent with one another. Please clarify this point.

- $\Delta\Delta E$ is not explained elsewhere in the paper and the figure isn't even labeled with this metric. I also could not find it mentioned in the original GEMME paper:
<https://academic.oup.com/mbe/article/36/11/2604/5548199>

-“We found that the small allosteric Parkin modulator BIO-2007817 was unable to confer significantly increased Parkin variant abundance.” It would be nice if there was a hypothesis presented as to why this is the case.

Minor changes:

-An x-axis label in fig. S10 is cut off. The legend also needs more detail.

-Figure 6D: why not label the two points that are R33Q and P437L?

-last paragraph before discussion “evolutionary conserved”  “evolutionarily conserved”

-“albeit the mature mRNA only spans 4 kb” rephrase for grammatical correctness.

-“However, likely most of the low abundance Parkin variants are targets of the PQC system independently of auto-ubiquitination.”  It is likely, however, that most of the low abundance Parkin variants are targets of the PQC system independent of auto-ubiquitination.

Reviewer #3 (Remarks to the Author):

The authors have generated a quantitative abundance map of nearly all 9300 possible Parkin missense and nonsense variants. They show that most low abundance variants are located in the structured domains of the protein, degraded by the proteasome and stabilized at lower temperature. They also provide a detailed, systematic analysis of degrons present in Parkin.

This is a beautiful and very interesting manuscript. The methodological approach is thorough and elegant. The manuscript is clearly written, even for non-experts in structural or computational biology. Parkin plays a crucial role in the pathogenesis of Parkinson’s disease, and a very large number of pathogenic variants as well as variants of unknown significance in the Parkin gene have been reported. The data and insights in this manuscript will be useful to refine predictions about the pathogenic versus benign consequences of these variants. The manuscript is therefore also clinically relevant.

I have only two minor other comments:

1. In the description of Fig. 2A the authors call some of the amino acid substitutions ‘synonymous’. This is a bit confusing, because a ‘synonymous’ variant usually means a variant that does not change the amino acid sequence. What do they mean here by synonymous?

2. In the second sentence of the Discussion, the authors state that 15% of PD cases are monogenic. This number is an overestimation. In most populations less than 5% of PD patients have an identifiable monogenic cause (e.g. PMID: 31324919). The percentage of 'genetic' cases rises to 10-15% when including patient with GBA mutations, but GBA mutations are genetic risk factors for PD rather than monogenic causes.

Reviewer #4 (Remarks to the Author):

In this manuscript, Clausen, Voutsinos, et al., studied nearly all individual missense mutational effects of Parkin on its stability using VAMP-seq. To unveil the biophysical mechanisms, they also performed experiments in various conditions including at altered temperatures, with small molecules and other perturbations. The highlight of their work is comparing mutational effects with the degtron-tiles where they show that mutations in the disordered regions likely strengthen the degtron potential. Overall, the generated data is comprehensive and meaningful, yet the data analysis is not thorough enough and needs improvement.

Here are specific comments:

(1) Mutations decrease protein stability by degtron exposure (structured parts) or strengthening degtron potential (disordered region): To strengthen the claim, authors should show how mutational effects in at least one tile of the structured region of the protein compared to the VAMP-seq result. One would expect it to show little correlation or a much more reduced correlation.

(2) It would be good to have an estimation of how much degtron creation/destruction or modification contributes to VAMP-seq abundance scores – either experimentally or via computational prediction.

(3) Authors should reword the section using Rosetta, as it is misleading. For example, "This indicates that the measured abundances are largely, but not fully, captured by the thermodynamic stability of the Parkin variants." This is wrong. The effects could be almost entirely due to stability changes - Rosetta is an imperfect predictor of stability ($R \sim 0.5$ is very consistent with many evaluations of how well Rosetta predicts stability, Tsuboyama, K. et al, Nature 2023; and various other mutational scanning papers with protein stability as the phenotypic output compared to Rosetta predictions or FoldX predictions).

(4) Pathological vs benign clinical variant discrimination: authors should compare their experimental data to state-of-the-art variant effect predictors like EVE (Frazer et al., Nature 2021).

(5) In various parts of the manuscript, reporting of the data analysis should be improved, with proper statistical tests and direct comparisons of the effects using scatterplots – not just distributions/patterns. For example, use more direct comparisons of the data in Figure to support the claims made in the text,

and report the correlation coefficients and p-values. This will make it easier for readers to interpret the results and understand the conclusions drawn from them. In addition, how do protease inhibition, PSMD14 siRNA, and 29-degree effects correlate? There is potentially insightful information to be drawn from thorough comparisons between mutational effects in these conditions rather than a brief description of 'slight' differences etc.

Minor points:

(6) Classifying mutations: Combining GeMME with stability assays, the authors classified mutations into three classes (class 1, wt-like 61%, class 2- unstable & nonfictional 21%, and stable but non-functional 15%). Authors could comment on the characteristics of these class-3 mutations. For instance, Are they at or close to the active sites? Also, the authors should clarify why 3% of mutations are not classified (61%+21%+15%=97%).

(7) Authors should mention if the six-disease-linked Parkin variants with WT-like abundance are accurately captured as stable but non-functional variants from the GeMME+ stability assay.

(8) Where are 24-tiles' breakpoints (every 12AA) located in a 3D-structural cartoon view? I wonder which of the tiles breaks a helix or a beta sheet secondary structures and whether they correlate with the stability.

(9) Throughout the manuscript, the authors should show SASA (solvent-accessible surface area) as a measurement of surface exposure information, rather than solely reporting on WCN.

(10) Figure S1 legend should include information on the three Parkin bands as to what they correspond.

(11) Figures S4 and S5 colour keys are missing.

(12) Wrong to count wildtype as one possible single amino acid substitution in "... abundance of 8757 out of 8836 (465 residues x 19 amino acid substitutions per position + 1 wildtype) possible single amino acid substitutions."

(13) It is not clear how many amino acid substitutions per DNA variant has in general. Based on what the authors state ("Reads comprising ten or more DNA substitutions or any indels were removed"), variants with up to 9 DNA substitutions are retained, which means there can be up to 3 amino acid substitutions (9/3=3, in case of 9 nucleotide substitutions). They should include information on the amino acid substitution distributions as a supplementary figure or table.

(14) The variant library (saturation mutagenesis library design) as nucleotide sequences should be made available (provided as a supplementary table for instance).

Our point-by-point response:

Reviewer #1 (Remarks to the Author):

It is established that numerous types of parkin mutation cause parkin-related PD. Recently, several studies have indicated that many disease-linked point mutants of Parkin retain substantial catalytic activity suggesting other mechanisms contributing to loss of protein function and hence disease. Here, in this very interesting piece of work by Clausen et al, the authors aimed to understand how mutations in Parkin contribute to its dysfunction, by systematically defining destabilizing Parkin mutations. Given the link between PRKN and PD, this has therapeutic application as increasing Parkin protein levels by inhibiting its degradation may restore the protein function and reverse the disease.

The authors generated a comprehensive Parkin variant library and used Vamp-seq to assign stability scores to every possible variant. Although most mutation does not affect protein stability, some mutations showed significant lower stability. Experiments were complemented by computational approaches to calculate thermodynamic stability and evolutionary conservation in order to pinpoint residues that are preserved to maintain a stable protein structure. Altogether, the authors suggest that destabilizing Parkin mutations expose PQC degrons that lead to Parkin proteolysis. To map PQC degrons that might be exposed because of the Parkin mutation the authors screened a tiled peptide library of Parkin.

-Overall, the authors have done a good job of identifying destabilizing Parkin mutations and potential PQC degron. The manuscript writing is clear and concise and figures are presented nicely. This resource should be of great interest to the readership of Nature communications both for investigators from the UPS and proteostasis field as well as for wider scientific community.

My main concern is that authors did not assess the functional consequence of the destabilizing Parkin mutations; some of the destabilized mutants may still be functional, even though they are partially degraded. The E3 ligase activity, localization and biological function (e.g mitophagy) were not analyzed for low abundance Parkin mutants and as such it is not clear whether the reduced abundance overall affects the protein function.

Our response:

We thank the reviewer for this thorough and positive evaluation of our work.

Major points:

1. Most PRKN variants' stability is similar to WT, while a small bump of unstable population exists. PSI calculation to my understanding does not take into account the fact that FACS bins are not sized equally and the PSI difference between bin1 to 2 is not the same as bin2 to 3. Can authors comment on that?

Our response:

The FACS machine is set to give the same number of cells in all bins which results in a non-constant bin width (Fig. 1F). The absolute fluorescence may change between settings and equipment, and it is in general a more robust strategy to rely on the fluorescence distribution rather than absolute fluorescence. The described issue arises from the unevenly spaced FACS gates (Fig. 1F), and absolute fluorescence does not necessarily fix this. We instead follow what has now become the standard in this field by using the PSI measure; this is based on the quantiles of the distribution, and we found this to be the most robust measure. In general, non-parametric measures are widely used, e.g. the Spearman correlation, and the PSI measure has, to our knowledge, been used in all previous applications of VAMP-seq (PMID: 29785012; PMID: 32094176; PMID: 34649609; PMID: 34314704) and most sort-seq based degron screens (PMID: 29779948; PMID: 36481666). We have included a summary of this discussion in the methods section (p.34).

The small shoulder of highly unstable variants in bin 1 (Fig. 1F) indicates that the resolution between variants of low and very low abundance is not high. This issue would have been much more pronounced if the library was expressed at lower level (please see our response to point 4 below).

2. Following up on the previous comment- can the authors create a figure similar to Fig S12 for variants enriched in bin1 only (those with PSIs 1-2, excluding stop mutants)? I wonder if the very low abundant variants cluster in specific protein domain or spread everywhere.

Our response:

This is a good suggestion that could reveal if the individual Parkin domains display a difference in their tolerance to mutations. We added a figure displaying the positioning of the very low median abundance variants (score <0.1) (excluding nonsense variant) in the supplemental material (Fig. S7). This revealed that these are found throughout the structured domains of Parkin. This result is mentioned in the manuscript on p.9.

3. Authors need to show that reduced Parkin levels lead to reduced protein activity in biological context- mitophagy assays and/or E3 ligase activity. Otherwise, correlation between destabilizing mutations and disease cannot be made. It could be that reducing the protein levels by 10-fold does not have not a major impact on cellular functions related to it. As there are only several rare pathogenic disease associated variants of low abundance (Fig 6A), proof of concept can be done on those.

Our response:

This is an excellent suggestion, which may also highlight if the GFP-fusion (see point 4 below) causes any potential artefacts. We agree that while total enzyme activity depends linearly on the enzyme amount, the cellular consequences and phenotype may not, and may further depend on the details of the phenotype and expression system. This is one reason why we also focus on correlating the observed abundances directly with the clinical consequences to suggest that reduced abundance is likely a common mechanism for several *PRKN* loss of function mutations. Indeed, reduced abundance is a widespread mechanism for many genetic disorders as well as cancer-linked somatic mutations in tumor suppressor genes, and a mechanism already probed for certain Parkin variants. For instance, it has been shown that that the pathogenic R42P and V56E variants show a 1:1 reduction in activity and

amount (PMID: 30994895), indicative of the above mechanism. We now stress this point in the manuscript (p.7 & p.22).

To approach the question of Parkin activity experimentally, we generated plasmids for landing pad integration to produce selected GFP-fused Parkin variants. Downstream, we introduced an IRES followed by the mt-Keima reporter commonly used for quantification of mitophagy (PMID: 21867919; PMID: 26549682). The new results are included in the revised manuscript (p.7 and Fig. S1), and show that without integration of Parkin, the HEK293T cells hardly induce mitophagy upon treating with antimycin/oligomycin. As expected, a catalytically dead (C431A) Parkin variant also fails to induce mitophagy, whereas WT Parkin efficiently induced mitophagy which can be blocked with bafilomycin. Our GFP-fused WT Parkin is therefore both stable and functional. Finally, and in agreement with previous reports (PMID: 30994895; PMID: 16714300; PMID: 15606901), we find that the low abundance pathogenic R42P variant is active. The fact that R42P appears as active wild-type Parkin is likely a consequence of it being overproduced in the cells. This shows that R42P is unstable but active, and its abundance is therefore a critical parameter for its pathogenicity. Accordingly, we now elaborate on this point in the manuscript (p.7 & p.22).

4. Expression from landing pad seems to be extremely high (Fig S1). To validate the Vamp-seq screens findings and exclude artifacts from overexpression and GFP-fusion, a few representative mutants need to be expressed under a weak promoter and protein levels need to be assessed in western blot. Parkin was reported be expressed endogenously in HEK293 cells so I would recommend comparing endogenous Parkin to exogenously variants in these cells.

Our response:

Although Parkin has been reported to be expressed in HEK293T cells we have never been able to detect endogenous Parkin by Western blotting (Fig. S1D), and we are therefore not able to perform the suggested experiment. RNA sequencing has also shown that *PRKN* mRNA in these cells is also extremely low (within the experiment noise) (<https://doi.org/10.1101/2023.10.02.560410>). These results are also consistent with recent work that shows high correlations across deep mutational scanning experiments across different expression levels, with the main effect being a change in the dynamical range, but overall full consistency at low and high levels (<https://doi.org/10.1126/sciadv.add9109>). As our abundance data correlate with conservation and thermodynamic stability (both predicted and determined *in vitro*), our results strongly suggest that although Parkin is overexpressed the abundance data are relevant also at lower expression levels. We now elaborate on this point in the discussion (p.22). However, certainly the Parkin expression level achieved from landing pad is likely to affect the dynamic range of the VAMP-seq. assay (see also our response to point 1 above). Thus, although we would have liked an even higher resolution, the high expression allows us to better discriminate between Parkin variants that are of low abundance. In our opinion this is a strength of our data, especially considering that the unstable pathogenic variants all display a very low abundance (Fig. 6B).

Although we cannot entirely exclude potential artefacts of the GFP fusion, the new activity data (Fig. S1) clearly shows that GFP-Parkin expressed from the landing pad is active upon mitophagy induction (please see our response to point 3 above). In addition, we note that although HEK293T cells may contain some undetectable level of endogenous Parkin, the mitophagy in cells without Parkin introduced in the landing pad is very low (Fig. S1C).

5. I am not sure what is the added value of the peptide screen. Hydrophobic degrons were discovered, however, we do not know which of them is exposed and serves as a PQC degnon in the context of Parkin low abundance variants. Do any of the Parkin variants is predicted to show severe structural changes and exposure of one of the discovered PQC degrons? Can authors provide an example? I am looking for a link between Parkin variant and peptide screens.

Our response:

The general hypothesis is that upon structural destabilization introduced by a missense mutation, one or several of the hydrophobic and buried degrons may be transiently exposed resulting in PQC-linked degradation. We have previously highlighted this mechanism for disease-linked variants in DHFR by mutagenesis coupled with structural studies by hydrogen/deuterium exchange and NMR spectroscopy (PMID: 35700725). As evident from that work, it is unfortunately not possible to predict which mutations will trigger the exposure of specific PQC degrons. However, the most likely scenario would be that mutations in a buried degnon region would lead to transient local unfolding and exposure of proximal PQC degrons, or at least degrons located within the same destabilized protein domain.

Our work supports the mechanism suggested above but adds a new (we believe interesting) angle: that missense variants may cause degradation via generating new, exposed degrons and not just causing exposure of existing (buried) degrons. Thus, the link between the Parkin variants and the degnon map is emphasized by our identification of the exposed region covered by tiles 9 and 10, where the effects of mutations introduced in the tiles and in full-length Parkin are strongly correlated (Fig. 4F). Hence, in this region of Parkin, missense variants may cause Parkin degradation without affecting the global structure at all. The new analyses added during the revision (Fig. S12), indicate that this is also the case in other regions of Parkin. In our opinion this provides important new mechanistic insights, and we now stress this point further in the manuscript (p.21).

Finally, we note the comments of Reviewer 4, who finds that the main strength of the work is our attempts at comparing mutational effects with degnon tiles.

Reviewer #2 (Remarks to the Author):

The authors present the generation and assaying of over 9000 single site mutants of the Parkin protein using the variant abundance by massively parallel sequencing approach. Using this method, they quantified the amount of Parkin, a ubiquitin-protein ligase implicated in Parkinson's disease, in cultured human cells and then did several analyses to get at the role of Parkin stability in proteostasis, mitophagy and the pathology of Parkinson's disease. They found that the abundance of Parkin in cells correlates with the thermostability of the variant as well as its propensity to be degraded. Abundance also correlated with Evolutionary Conservation according to the GEMME metric, and ½ of the disease-linked Parkin variants have low abundance. Nevertheless, the best metrics for predicting the disease-linked mutants were Rosetta $\Delta\Delta G$ and GEMME score, which the authors interpret to mean that the abundance score only captures some of the pathogenic attributes of Parkin variants.

Overall, the work is extensive enough and of broad enough interest both from a basic science and an applied science standpoint to warrant publication in a journal targeted to a wide audience. The report is also well-written and the conclusions mostly justified. There are only a few problems they could address to improve its presentation and support for their arguments. With these issues satisfactorily addressed, I see no reason not to publish these findings.

Our response:

We thank the reviewer for the encouraging evaluation and for spending time on our work.

- The analysis using GEMME is being discussed as if it is a black box that everyone knows what is going on in there. Their introduction to the method didn't specify that "evolutionary conservation" refers to a metric derived from a global model that considers the role of epistasis. The background on this method needs to be explained better.

Our response:

We thank the reviewer for pointing this out and we now elaborate on the GEMME tool (p.14).

-Next, figure S10 doesn't depict anything about low conservation based on the evolutionary conservation score. The statement "When comparing the abundance map with the GEMME map, regions with low conservation also generally appeared tolerant to mutations (Fig. S10)" is really hard to see in figure S10. Perhaps there is another way to demonstrate this point or to annotate the current figure or preferably to come up with a numerical analysis instead.

Our response:

We agree and have changed the text accordingly (p. 15). In regard, to the numerical analysis we refer to Fig. 5B where the abundance and GEMME scores are correlated directly.

-I have a problem with putting experimental data of thermostability in the supplemental (Fig. S11) instead of with the Rosetta analyses in the main text (Fig. 5). They should discuss the experimental thermostability data first and then the Rosetta predictions as a support, rather than the other way around. You should show the correlation of Rosetta $\Delta\Delta G$'s with the experimental ones from the 35 that were tested, and use that to bolster trust that Rosetta performs well in this system.

Our response:

As suggested, we now plot the Rosetta predicted $\Delta\Delta G$ s vs. the experimentally determined melting temperatures from Stevens et al. (Fig. S14). The correlation coefficient (-0.54) is in line with, though perhaps slightly lower than, the values previously observed in benchmarks for Rosetta stability predictions (for example in <https://doi.org/10.1021/acs.jctc.6b00819>, <https://doi.org/10.1016/j.csbj.2022.11.048> and <https://doi.org/10.1093/bib/bbab555>). Differences between the predictions and experiments might be explained for example by remaining inaccuracies in Rosetta, and the fact that the experiments probe changes in apparent melting temperature and not thermodynamic stability per se). Further, we note that the relationship between changes in thermodynamic stability ($\Delta\Delta G$) and melting temperature (even when the protein folds reversibly)

depends on the size of the protein (<https://doi.org/10.1021/acs.jpcb.7b05684>). Thus, if individual domains in Parkin would unfold individually, one would not expect a perfect correlation between experimental values of $\Delta\Delta G$ and ΔT_m . Presumably Rosetta will be most robust at capturing very unstable Parkin variants, that could be so unstable that purifying the proteins and probing the melting temperatures would be impossible. Thus, in our opinion this point relates to that raised by Reviewer #4 (point 3). We have modified the text accordingly (p.14), but we have not reorganized the order by which the results are presented.

-“For the remaining six variants, four are evolutionary [sic] conserved($\Delta\Delta E < -2$), indicating a connection to Parkin function.”

This statement claims that evolutionary conservation increases the smaller the $\Delta\Delta E$ value is. But authors state on page 14, “Using GEMME, we calculated an evolutionary conservation score, where a score of zero predicts a WT-like variant, while variants that are predicted as unfavorable for structure and/or function result in a negative score.” I am confused as to how these two statements are consistent with one another. Please clarify this point.

Our response:

We apologize for the text being unclear here. Negative GEMME scores indicate that the substitution is not compatible with the alignments, whereas a GEMME score of zero indicates that the substitution is compatible with the alignments and therefore should be neutral/allowed (or WT-like in terms of function and stability). In the case where most changes are incompatible with alignment, i.e. have negative values, we say that the wild-type amino acid is conserved. We have modified the text accordingly (p.15). Hence, GEMME scores < -2 correctly indicate that the residues are conserved for functional and/or structural reasons.

$-\Delta\Delta E$ is not explained elsewhere in the paper and the figure isn't even labeled with this metric. I also could not find it mentioned in the original GEMME paper:

<https://academic.oup.com/mbe/article/36/11/2604/5548199>

Our response:

We apologize, $\Delta\Delta E$ is our internal name for GEMME scores. We have removed this in the revised manuscript.

-“We found that the small allosteric Parkin modulator BIO-2007817 was unable to confer significantly increased Parkin variant abundance.” It would be nice if there was a hypothesis presented as to why this is the case.

Our response:

We now elaborate on this point, namely that binding of a small molecule to the folded state of Parkin might be able to increase the population of the folded protein (p. 11 & p.22-23).

Minor changes:

-An x-axis label in fig. S10 is cut off. The legend also needs more detail.

Our response:

We corrected this and expanded the legend (Fig. S13).

-Figure 6D: why not label the two points that are R33Q and P437L?

Our response:

We have labelled the points as suggested (Fig. 6D).

-last paragraph before discussion “evolutionary conserved”  “evolutionarily conserved”

Our response:

Thank you. We have corrected the mistake (p.18).

-“albeit the mature mRNA only spans 4 kb” rephrase for grammatical correctness.

Our response:

We reworded this (p.19).

-“However, likely most of the low abundance Parkin variants are targets of the PQC system independently of auto-ubiquitination.”  It is likely, however, that most of the low abundance Parkin variants are targets of the PQC system independent of auto-ubiquitination.

Our response:

Thank you. We corrected the text accordingly (p.20).

Reviewer #3 (Remarks to the Author):

The authors have generated a quantitative abundance map of nearly all 9300 possible Parkin missense and nonsense variants. They show that most low abundance variants are located in the structured domains of the protein, degraded by the proteasome and stabilized at lower temperature. They also provide a detailed, systematic analysis of degrons present in Parkin.

This is a beautiful and very interesting manuscript. The methodological approach is thorough and elegant. The manuscript is clearly written, even for non-experts in structural or computational biology. Parkin plays a crucial role in the pathogenesis of Parkinson’s disease, and a very large number of pathogenic variants as well as variants of unknown significance in the Parkin gene have been reported. The data and insights in this manuscript will be useful to refine predictions about the

pathogenic versus benign consequences of these variants. The manuscript is therefore also clinically relevant.

Our response:

We thank the reviewer for the kind words and input to our manuscript.

I have only two minor other comments:

1. In the description of Fig. 2A the authors call some of the amino acid substitutions 'synonymous'. This is a bit confusing, because a 'synonymous' variant usually means a variant that does not change the amino acid sequence. What do they mean here by synonymous?

Our response:

Indeed, we realize that this was not clear. By synonymous variants we mean what used be termed silent mutations, i.e. where a codon is exchanged with a synonymous one (encoding the same amino acid). Those variants with abundance scores similar to the synonymous wild-type variants are colored white in the heat-map and obtain a synonymous score. We now state this in the results section (p.8) and the legend to Fig. 2.

2. In the second sentence of the Discussion, the authors state that 15% of PD cases are monogenic. This number is an overestimation. In most populations less than 5% of PD patients have an identifiable monogenic cause (e.g. PMID: 31324919). The percentage of 'genetic' cases rises to 10-15% when including patient with GBA mutations, but GBA mutations are genetic risk factors for PD rather than monogenic causes.

Our response:

We thank the reviewer for pointing this out and we have corrected the text accordingly (p.19).

Reviewer #4 (Remarks to the Author):

In this manuscript, Clausen, Voutsinos, et al., studied nearly all individual missense mutational effects of Parkin on its stability using VAMP-seq. To unveil the biophysical mechanisms, they also performed experiments in various conditions including at altered temperatures, with small molecules and other perturbations. The highlight of their work is comparing mutational effects with the degron-tiles where they show that mutations in the disordered regions likely strengthen the degron potential. Overall, the generated data is comprehensive and meaningful, yet the data analysis is not thorough enough and needs improvement.

Our response:

We are grateful for the reviewer spending time on providing valuable feedback on our work.

Here are specific comments:

(1) Mutations decrease protein stability by degron exposure (structured parts) or strengthening degron potential (disordered region): To strengthen the claim, authors should show how mutational effects in at least one tile of the structured region of the protein compared to the VAMP-seq result. One would expect it to show little correlation or a much more reduced correlation.

Our response:

The way we interpret this is that the reviewer wishes us to introduce mutations in a buried degron tile as a sort of control for the exposed region (tiles 9 & 10). To approach this, we selected tile 23 and introduced different variants into the tile. These new data revealed, as expected, a poor correlation between the degron potential of the tile variants and the abundance of the corresponding variants in full-length Parkin. The results are included as a new figure in the supplemental material (Fig. S11) and are mentioned in the manuscript (p.12).

(2) It would be good to have an estimation of how much degron creation/destruction or modification contributes to VAMP-seq abundance scores – either experimentally or via computational prediction.

Our response:

This is an excellent suggestion and, in our opinion, proved to be a very valuable addition to our work. We have included a new analysis of variants at solvent exposed positions that do not affect the structural stability substantially (as predicted by Rosetta). The analysis shows that, among variants in this set with low abundance, many also increase the degron potential suggesting that the mechanism of abundance loss is the introduction of a degradation patch (degron) in an already exposed region, including the ACT region already discussed. These new data are included in the supplemental material (Fig. S12) and are discussed in manuscript (p.14 & p.21).

(3) Authors should reword the section using Rosetta, as it is misleading. For example, "This indicates that the measured abundances are largely, but not fully, captured by the thermodynamic stability of the Parkin variants." This is wrong. The effects could be almost entirely due to stability changes - Rosetta is an imperfect predictor of stability ($R \sim 0.5$ is very consistent with many evaluations of how well Rosetta predicts stability, Tsuboyama, K. et al, Nature 2023; and various other mutational scanning papers with protein stability as the phenotypic output compared to Rosetta predictions or FoldX predictions).

Our response:

We agree on this important issue, and we thank the reviewer for pointing it out. We have modified the text accordingly (p.14).

(4) Pathological vs benign clinical variant discrimination: authors should compare their experimental data to state-of-the-art variant effect predictors like EVE (Frazer et al., Nature 2021).

Our response:

We added a ROC curve for the EVE predictions (Fig. 6C) and include correlations with our experimental data (Fig. S17A) and with GEMME (Fig. S17B). Indeed, EVE outperforms GEMME, Rosetta and the abundance scores for distinguishing pathogenic from non-pathogenic variants, and correlates with our experimental observations. We now mention this in the manuscript (p.18).

(5) In various parts of the manuscript, reporting of the data analysis should be improved, with proper statistical tests and direct comparisons of the effects using scatterplots – not just distributions/patterns. For example, use more direct comparisons of the data in Figure to support the claims made in the text, and report the correlation coefficients and p-values. This will make it easier for readers to interpret the results and understand the conclusions drawn from them. In addition, how do protease inhibition, PSMD14 siRNA, and 29-degree effects correlate? There is potentially insightful information to be drawn from thorough comparisons between mutational effects in these conditions rather than a brief description of ‘slight’ differences etc.

Our response:

We have strived to improve the data analyses in the paper. However, we believe that reviewer has misunderstood the results presented from the perturbation experiments in Fig. 3. We only sorted cells and performed the full VAMP seq. analyses for unperturbed cells (no inhibitors, 37 C). Although, potentially interesting, it is therefore not possible for us to correlate the results from the different perturbations. The purpose of the data presented in Fig. 3 is simply to reveal (on a global scale) that for most variants the low abundance is dependent on temperature and the proteasome.

Minor points:

(6) Classifying mutations: Combining GeMME with stability assays, the authors classified mutations into three classes (class 1, wt-like 61%, class 2- unstable & nonfictional 21%, and stable but non-functional 15%). Authors could comment on the characteristics of these class-3 mutations. For instance, Are they at or close to the active sites? Also, the authors should clarify why 3% of mutations are not classified ($61\%+21\%+15\%=97\%$).

Our response:

We have expanded the text on the functional classification (p.16) and included an additional supplemental figure (Fig. S15) to explain the distinctions more clearly.

(7) Authors should mention if the six-disease-linked Parkin variants with WT-like abundance are accurately captured as stable but non-functional variants from the GeMME+ stability assay.

Our response:

We now mention this in the text (p. 18).

(8) Where are 24-tiles' breakpoints (every 12AA) located in a 3D-structural cartoon view? I wonder which of the tiles breaks a helix or a beta sheet secondary structures and whether they correlate with the stability.

Our response:

The purpose of the tiling experiments is to uncouple protein sequence from structure, and in this way force the exposure of regions that would normally be buried within the native fold to test their degron potential. As all the included tiles are 24 residues long, we believe that there is either no or only very limited (transient/unstable) secondary structure in the peptides. Accordingly, we do not expect by this approach to be able to identify any structured degrons. We have therefore not further analyzed how structure may potentially impact the degron potential.

(9) Throughout the manuscript, the authors should show SASA (solvent-accessible surface area) as a measurement of surface exposure information, rather than solely reporting on WCN.

Our response:

We now include accessible surface area as a metric (Fig. S6 and supplemental Excel spreadsheet). As this metric and WCN (as expected) correlate inversely (Fig. S6) we did not change the main figures.

(10) Figure S1 legend should include information on the three Parkin bands as to what they correspond.

Our response:

The major band corresponds to full-length GFP-Parkin. We have included a new blot which we hope is more clear (Fig. S1D).

(11) Figures S4 and S5 colour keys are missing.

Our response:

The information is given in the figure legends, and we now added a color key to the figures (Fig. S4 and S5).

(12) Wrong to count wildtype as one possible single amino acid substitution in "... abundance of 8757 out of 8836 (465 residues x 19 amino acid substitutions per position + 1 wildtype) possible single amino acid substitutions."

Our response:

We have corrected the text (p.8).

(13) It is not clear how many amino acid substitutions per DNA variant has in general. Based on what the authors state (“Reads comprising ten or more DNA substitutions or any indels were removed”), variants with up to 9 DNA substitutions are retained, which means there can be up to 3 amino acid substitutions ($9/3=3$, in case of 9 nucleotide substitutions). They should include information on the amino acid substitution distributions as a supplementary figure or table.

Our response:

These numbers are now summarized in the supplemental material (Fig. S18).

(14) The variant library (saturation mutagenesis library design) as nucleotide sequences should be made available (provided as a supplementary table for instance).

Our response:

The previously released barcode map only listed the amino acid variants, so we have released a new file on GitHub ([pacbio/variants_syn_single.csv.gz](https://github.com/pacbio/variants_syn_single.csv.gz)), that explicitly lists all 249,971 nucleotide variants of WT, synonymous WT, and single amino acid variants.

REVIEWERS' COMMENTS

Reviewer #1 (Remarks to the Author):

The authors have satisfactorily addressed all questions and concerns raised from this review, and as such is suitable for publication.

Reviewer #4 (Remarks to the Author):

I've thoroughly reviewed the revised manuscript and appreciate the efforts made in addressing the raised points. The authors addressed all my raised points satisfactorily. Especially, the analysis in response to the original point 2 is very nice, bringing in new insights from Figure S12A.

I have a few suggestions and corrections:

1. Regarding the original point 2 analysed in Figure S12A:

- Please report the number of total data points in Figure S12A, as the reported 63 is only a small subset of the data points shown above.
- Please report the correlation coefficient for the negative correlation between QCDPred and abundance scores. This could further support the significance of degron creation/destruction.

2. Regarding the original point 4 EVE scores: Please include details on how EVE scores are calculated in the Methods section since it has been calculated and included in the revised manuscript.

3. Errors and Missing Information:

- In Figure 4E, the Spearman correlation (0.79) should be -0.79.
- In Figure S17 Panel A, the Spearman correlation (0.61) should be -0.61.
- Provide missing Spearman correlation coefficients in S6 and S11. Especially for S11, explicitly reporting the correlation coefficient to directly compare with the reported Spearman's rho 0.86 (Figure 4F, page 12) would enhance the credibility.
- Correct on Page 23, “, our cDNA library ...” to “Our cDNA library ...”

4. I suggest modifying the legend for Figure S12 to clarify the reference to "score < 0.6; 44% or 1% of all low-abundance variants", as it is not immediately obvious which 44% refers to. For instance, it would be better to clarify it like "score < 0.6; 44% of the 63 variants or 1% of all low-abundance variants"

5. Ensure consistency in Spearman's rho annotation throughout the figure panels ('r', 'ρ', and 'Spearman's' throughout the figures are used).

Our point-by-point response

Reviewer #1 (Remarks to the Author):

The authors have satisfactorily addressed all questions and concerns raised from this review, and as such is suitable for publication.

Our response:

We thank the reviewer for their valuable time and input to our work.

Reviewer #4 (Remarks to the Author):

I've thoroughly reviewed the revised manuscript and appreciate the efforts made in addressing the raised points. The authors addressed all my raised points satisfactorily. Especially, the analysis in response to the original point 2 is very nice, bringing in new insights from Figure S12A.

Our response:

We thank the reviewer for their valuable time and input to our work.

I have a few suggestions and corrections:

1. Regarding the original point 2 analysed in Figure S12A:

- Please report the number of total data points in Figure S12A, as the reported 63 is only a small subset of the data points shown above.
- Please report the correlation coefficient for the negative correlation between QCDPred and abundance scores. This could further support the significance of degran creation/destruction.

Our response:

We included this information in the legend to supplementary figure 12.

2. Regarding the original point 4 EVE scores: Please include details on how EVE scores are calculated in the Methods section since it has been calculated and included in the revised manuscript.

Our response:

The EVE scores were not calculated, but downloaded directly from the EVE site (<https://evemodel.org/>). We now state this in the methods section (p38).

3. Errors and Missing Information:

- In Figure 4E, the Spearman correlation (0.79) should be -0.79.
- In Figure S17 Panel A, the Spearman correlation (0.61) should be -0.61.
- Provide missing Spearman correlation coefficients in S6 and S11. Especially for S11, explicitly reporting the correlation coefficient to directly compare with the reported Spearman's rho 0.86 (Figure 4F, page 12) would enhance the credibility.
- Correct on Page 23, “, our cDNA library ...” to “Our cDNA library ...”

Our response:

We have made these corrections, except for the one on p23, which is not a typo.

4. I suggest modifying the legend for Figure S12 to clarify the reference to "score < 0.6; 44% or 1% of all low-abundance variants", as it is not immediately obvious which 44% refers to. For instance, it would be better to clarify it like "score < 0.6; 44% of the 63 variants or 1% of all low-abundance variants"

Our response:

Thank you, we have changed the text to increase clarity.

5. Ensure consistency in Spearman's rho annotation throughout the figure panels ('r', 'ρ', and 'Spearman's' throughout the figures are used).

Our response:

We have made these corrections.